# Thermodynamically stable whilst kinetically labile coordination bonds lead to strong and tough self-healing polymers

Jian-Cheng Lai [1], Xiao-Yong Jia[1], Da-Peng Wang[1], Yi-Bing Deng[1], Peng Zheng [1], Cheng-Hui Li[1], Jing-Lin Zuo [1] & Zhenan Bao [2]

There is often a trade-off between mechanical properties (modulus and toughness) and dynamic self-healing. Here we report the design and synthesis of a polymer containing thermodynamically stable whilst kinetically labile coordination complex to address this conundrum. The Zn-Hbimcp (Hbimcp = 2,6-bis((imino)methyl)-4-chlorophenol) coordination bond used in this work has a relatively large association constant ($2.2 \times 10^{11}$) but also undergoes fast and reversible intra- and inter-molecular ligand exchange processes. The as-prepared **Zn(Hbimcp)$_2$-PDMS** polymer is highly stretchable (up to 2400% strain) with a high toughness of 29.3 MJ m$^{-3}$, and can autonomously self-heal at room temperature. Control experiments showed that the optimal combination of its bond strength and bond dynamics is responsible for the material's mechanical toughness and self-healing property. This molecular design concept points out a promising direction for the preparation of self-healing polymers with excellent mechanical properties. We further show this type of polymer can be potentially used as energy absorbing material.

[1] State Key Laboratory of Coordination Chemistry, School of Chemistry and Chemical Engineering, Nanjing National Laboratory of Microstructures, Collaborative Innovation Center of Advanced Microstructures, Nanjing University, Nanjing 210093, People's Republic of China. [2] Department of Chemical Engineering, Stanford University, Stanford, CA 94305, USA. Correspondence and requests for materials should be addressed to C.-H.L. (email: chli@nju.edu.cn) or to J.-L.Z. (email: zuojl@nju.edu.cn) or to Z.B. (email: zbao@stanford.edu)

Self-healing materials have received extensive attention during the past decade[1–4]. A number of strategies, including encapsulation of healing agents and utilization of reversible chemical bonds, have been adopted to design self-healing materials[5–18]. For most self-healing materials based on dynamic crosslinked networks, there is often a trade-off between mechanical properties (modulus and toughness) and dynamic healing, i.e., stronger interactions often result in tough modulus but less dynamic systems, slowing down autonomous healing, while weaker interactions afford faster self-healing, but yield soft and viscoelastic materials[19,20]. However, the practical application of self-healing materials generally requires a combination of high toughness and autonomous self-healing.

Several concepts have been proposed to address this conundrum[19,21–24]. Guan and colleagues[19,21] used a multiphase strategy for self-healing materials design, in which the hard phase provides stiffness and strength to the material while the multi-valent supramolecular interactions in the soft matrix enable autonomous self-healing. However, the mechanical properties of such multicomponent systems depend on the detailed aggregation morphology, and potential loss of structural order may occur depending on processing conditions. Suo and colleagues[22] reported a series of double-network gels with high toughness and self-healing property. However, those hydrogels tend to have low Young's modulus below 0.1 MPa, which are desirable for tissue engineering, but unfavorable for applications where load-bearing is required. We previously reported a strategy to circumvent the above issues by tuning the polymer chain conformation and bond strength of metal–ligand interactions[12,25,26]. We reported a multi-strength metal–ligand system with both strong and weak metal–ligand binding sites adjacent to each other, which gave rise to highly stretchable and self-healing elastomers. However, our materials also had low toughness and low Young's modules below 0.5 MPa. Recently, we also reported a rigid and healable polymer, which was crosslinked by weak (with an association constant of $4.10 \times 10^4 \, M^{-1}$) but abundant Zn(II)–carboxylate interactions. However, this polymer shows low fracture strain and cannot be healed at room temperature[27].

The mechanical properties of a polymer with similar polymer structures are determined by the thermodynamic stability of the crosslinking sites[28,29]. The more stable (i.e., higher association constant) of the crosslinking sites, the stronger and tougher but less dynamic of the polymer. In contrast, the self-healing rate of a polymer is determined by the kinetic lability of the crosslinking sites. Therefore, to achieve both high toughness/high modulus while having an autonomous self-healing property, a molecular design concept for the crosslinking site that is thermodynamically stable while kinetically labile is needed. However, this is not readily achievable in common molecular systems.

We turn to coordination bonds as they consist of unique non-covalent interactions between a metal ion and its surrounding organic ligands. The strength of such interactions is highly tunable. With different combinations of metal ions and ligands, the bond strength can be readily adjusted in a broad range from approximately 25 to 95% of a covalent C–C bond (with a bond energy of about 350 kJ mol$^{-1}$). Moreover, many coordination complexes exhibit intra- and inter-molecular bond exchange through dissociative and associative processes. Despite their highly dynamic nature, they are thermodynamically stable[28,29]. It is therefore possible to obtain mechanically robust (strong and tough) self-healing polymers through fine tuning the thermodynamic as well as kinetic properties of coordination bonds, which cannot be easily achieved through designing of covalent bonds (such as disulfide bonds) or non-covalent interactions (such as hydrogen bonds).

To demonstrate the concept, we design and synthesize a poly (dimethylsiloxane) (PDMS) polymer containing an alterdentate ligand, 2,6-bis((propylimino)methyl)-4-chlorophenol (denoted as Hbimcp). This ligand can provide two equivalent imine-N donor centers. However, due to the steric hindrance, the two equivalent imine-N donor centers cannot coordinate with the same metal ion (Zn(II) in our study) at the same time and therefore the two coordination atoms are alternative and interchangeable. On the other hand, Zn(II)-2,6-bis((propylimino)methyl)-4-chlorophenol (denoted as **Zn(Pr-Hbimcp)$_2$**) complex has a relatively large association constant ($2.2 \times 10^{11} \, M^{-1}$). Such a unique coordination system would be an ideal crosslinking site that is both thermodynamically stable and kinetically labile and can lead to strong and tough self-healing polymers. We incorporate this coordination system into PDMS to obtain a **Zn(Hbimcp)$_2$-PDMS** polymer. The **Zn(Hbimcp)$_2$-PDMS** polymer is highly stretchable up to 2400% strain with a high toughness of 29.3 MJ m$^{-3}$, and can autonomous self-heal at room temperature. Control experiments by varying the metal ions and metal-to-ligand molar ratios showed that the bond strength and bond dynamics play the key role in determining the material's mechanical and self-healing property. The design concept in this work may represent a general approach to the preparation of self-healing polymers with excellent mechanical properties, while the resulting polymer may find promising applications as energy absorbing materials for car crash protection, sportswear, shockproof pad, armored clothing, etc.

## Results

**Material design and characterizations**. Alterdentate ligands offer at least two equivalent sites when binding to metal ions. Their metal complexes have been studied several decades ago by Minkin and colleagues[30–33] and Von Zelewsky et al.[34,35]. In these complexes, the alternative donor centers cannot participate simultaneously in coordination due to steric hindrance. As a result, complexes with uncoordinated donor centers were formed. Since the alternative donor centers are equivalent in nature, exchange of one coordinated metal ion from one site to another can occur by inter- or intramolecular mechanisms. Inspired by these results, we selected a Zn(II)-2,6-bis((propylimino)methyl)-4-chlorophenol (denoted as Zn(Pr-Hbimcp)$_2$) and Zn(II)-2,6-bis((benzylimino) methyl)-4-chlorophenol (denoted as Zn(Bz-Hbimcp)$_2$) for our design. According to studies on model ligand 2,6-bis((propylimino)methyl)-4-chlorophenol (Pr-Hbimcp), various Zn-Hbimcp complexes, such as Zn(Hbimcp), Zn$_1$(Hbimcp)$_2$, Zn$_2$(Hbimcp)$_2$, Zn$_2$(Hbimcp)$_3$ and other unknown species, were formed as mixture upon reacting Pr-Hbimcp with ZnCl$_2$, with different content percentages when different amount of ZnCl$_2$ was added. With ligand-to-metal molar ratio of 2:1, the cationic complex [Zn(Pr-Hbimcp)$_2$]$^{2+}$, in which the Zn(II) ions were coordinated to two oxygen and two nitrogen atoms with four equivalent conformations[33] (Fig. 1a), has the maximum abundance in the electrospray ionization mass spectra (ESI-MS) (except for the free ligand fragments generated due to electrospray ionization, Supplementary Figure 1a). Upon decreasing the ligand-to-metal molar ratio to 1:1, the neutral complex Zn$_2$(bimcp)$_2$Cl$_2$ dominates gradually in the products (Supplementary Figure 2 and Supplementary Figure 3). The ESI-MS of the mixture of [Zn(Pr-Hbimcp)$_2$]$^{2+}$ (597.4) and [Zn(Bz-Hbimcp)$_2$]$^{2+}$(789.5) exhibits the peak of [Zn(Pr-Hbimcp)(Bz-Hbimcp)]$^{2+}$(693.4), indicating that inter-molecular ligand exchange occurred in the solutions (Supplementary Figure 1). Moreover, The $^1$H spectrum of Zn(Pr-Hbimcp)Cl$_2$ displayed high fluxionality in solution at room temperature as evidenced from the single spin doublet signals for propyl groups, indicating a rapid ligand exchange process at the Zn(II) center (Fig. 1b, Supplementary Figure 4). When the temperature was decreased to below 0 °C, the spectrum showed

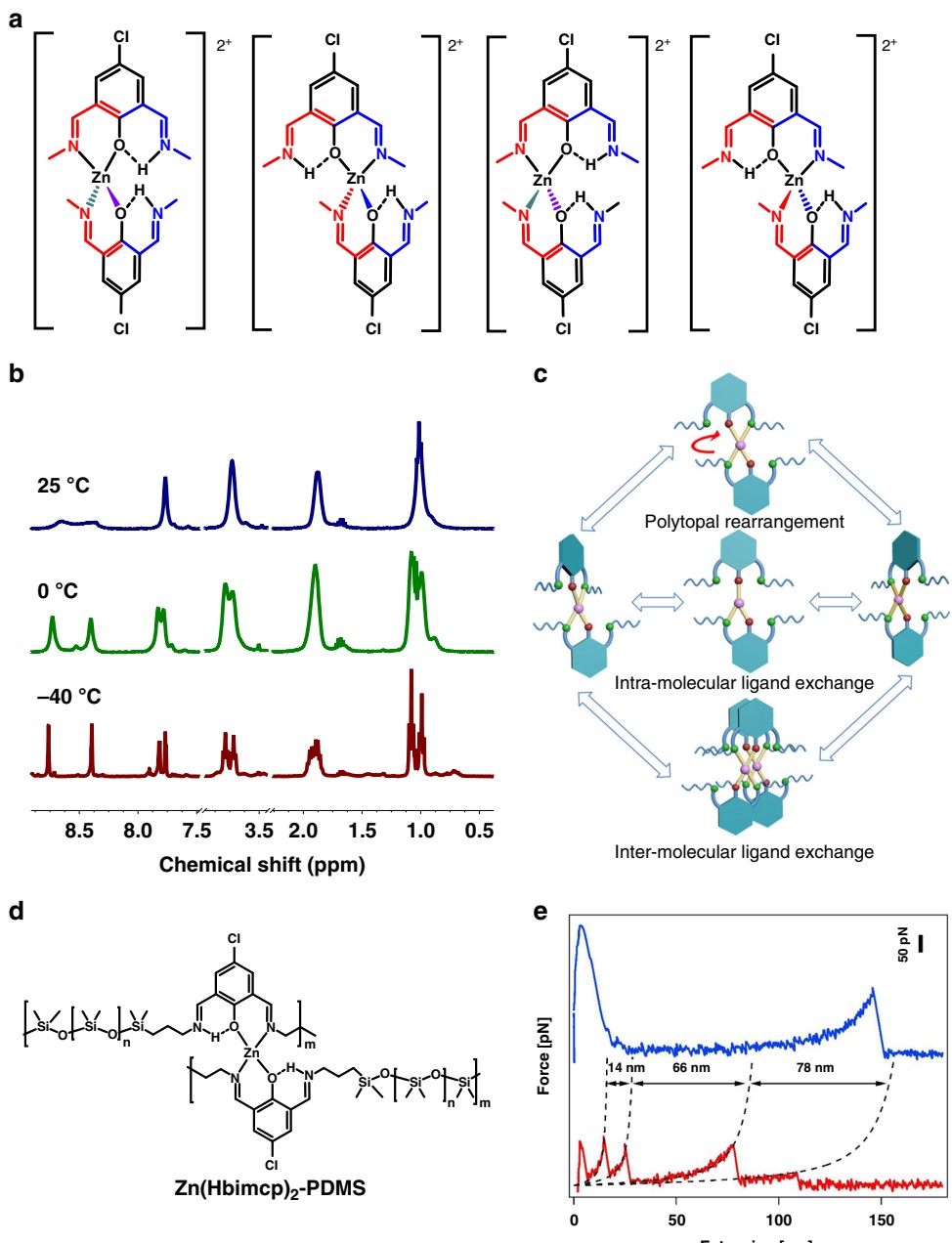

**Fig. 1** Structure and characterization of model complex and polymer. **a** Possible stereochemical structures for [Zn(Hbimcp)$_2$]$^{2+}$ complex. **b** Variable temperature $^1$H nuclear magnetic resonance (NMR) for Zn(Pr-Hbimcp)Cl$_2$. **c** Three possible pathways for ligand exchange process. **d** The structure of polymer complex **Zn(Hbimcp)$_2$-PDMS**. **e** Typical force–extension curves during force spectroscopy measurements of the stretching of a single chain of **Hbimcp-PDMS** (blue line) and of **Zn(Hbimcp)$_2$-PDMS** (red line). Scale bar for the vertical axis, 50 pN

two sets of signals due to the diastereotopic splitting (the protons of the chelated and non-chelated aldimine fragments are no longer equivalent), indicating that the ligand exchange process was frozen[36]. All these results confirm the highly dynamic nature of the Zn(II) complexes at room temperature. The ligand exchange process can be intra-molecular or inter-molecular through three possible pathways (Fig. 1c): (i) polytopal rearrangement of the diagonal twist type involving passage through a planar trans-structure; (ii) intramolecular process determined by the dissociation–reformation of one of the Zn(II)-N bonds; and (iii) intermolecular degenerate ligand exchange process.

The above dynamic Zn(II) complexes are then introduced into a linear PDMS polymer backbone, which is selected due to its broad

usages for stretchable electronics and good dielectric property[37,38] (Fig. 1d). Briefly, the PDMS oligomer containing 2,6-bis((propy-limino)methyl)-4-chlorophenol groups (**Hbimcp-PDMS**) were prepared by condensation reactions between bis(3-aminopropyl) terminated poly(dimethylsiloxane) (H$_2$N-PDMS-NH$_2$, $M_n$ = 700–900) and 5-chloro-2-hydroxyisophthalaldehyde to give an orange viscous liquid. It was subsequently crosslinked by addition of Zn(II) chloride, with a molar ratio of Zn(II) ion to Hbimcp ligand of 1:2, yielding a dark-red solid (**Zn(Hbimcp)$_2$-PDMS**, Supplementary Figure 5). The ultraviolet–visible (UV–Vis) spectrum of the thin film shows a band at 422 nm, as shown in Supplementary Figure 6, similar to the absorption wavelength observed for [Zn(Pr-Hbimcp)$_2$]$^{2+}$ (424 nm). Furthermore, the

intensity of the O–H stretching at 3454 cm$^{-1}$ partially decreased and an N–H stretching frequency at 3353 cm$^{-1}$ was observed in Fourier transform infrared (FTIR) spectra (Supplementary Figure 7). These results indicate the dominant presence of [Zn(Hbimcp)$_2$]$^{2+}$ complexes.

We used single-molecule force spectroscopy to characterize the dynamic nature of Zn-Hbimcp coordination bonds. As shown in Fig. 1e, stretching of **Hbimcp-PDMS** polymer chain results in only one force peak, which corresponds to the detachment of the macromolecule from the substrate. However, stretching of a **Zn(Hbimcp)$_2$-PDMS** single polymer chain leads to typical sawtooth-like force–extension curves, in which the different force peaks correspond to the unfolding of PDMS units through the rupture of coordination bonds in the [Zn(Hbimcp)$_2$]$^{2+}$ complexes. The contour length increments are quite similar to those for Fe-Hpdca-PDMS[12], although the molecular lengths of the repeating units (NH$_2$-PDMS-NH$_2$) are different. This is because the polymer chain in **Zn(Hbimcp)$_2$-PDMS** becomes quite rigid with the shorter repeating units. Only the bimcp ligands separated by several NH$_2$-PDMS-NH$_2$ repeating units can be connected through coordination to Zn(II) metal ions in the same polymer chain (Supplementary Figure 8). The average rupture force of the [Zn(Hbimcp)$_2$]$^{2+}$ coordination complexes, as determined from 1566 stretching of single-chain experiments (Supplementary Figure 9), is about 108.5 ± 40.9 pN (mean ± s.d., $n = 1566$). Moreover, reversible unfolding/refolding of **Zn(Hbimcp)$_2$-PDMS** was observed as shown in Supplementary Figure 10. These results indicated the single-chain **Zn(Hbimcp)$_2$-PDMS** molecule can be unfolded and refolded because of the rupture and reconstruction of [Zn(Hbimcp)$_2$]$^{2+}$ complexes.

**Mechanical and self-healing properties**. The glass transition temperature ($T_g$) for the resulting polymer network was measured to be below −23.1 °C (Supplementary Figure 11), which is typical for silicone rubbers. Temperature sweeping rheological test showed that both the storage modulus $G'$ and loss modulus $G''$ decreased upon increasing the temperature, but $G'$ decreased more rapidly than $G''$ (Supplementary Figure 12). The storage modulus is lower than the loss modulus at temperature higher than 50 °C, indicating the enhanced mobility of the polymer upon heating. The time–temperature superposition (TTS) of rheological data at 25 °C showed that $G'$ is lower than $G''$ at low frequencies and higher at high frequency (Fig. 2a). Figure 2b shows the temperature-dependent characteristic relaxation time of the polymer calculated from the frequency of the intersection point in TTS curves at different temperature. The characteristic relaxation time at −25 °C is as long as 10$^{10}$ s but drops exponentially to 8 s at 25 °C. Such results correlate well with the fact that the ligand exchange process of the Zn(Pr-Hbimcp)Cl$_2$ complex was observed at 25 °C but not at −40 °C in the variable temperature $^1$H nuclear magnetic resonance (NMR) measurement. Therefore, the crosslinking interactions in the **Zn(Hbimcp)$_2$-PDMS** polymer are highly dynamic at room temperature, thus facilitating the autonomous self-healing.

The obtained **Zn(Hbimcp)$_2$-PDMS** polymer exhibited high Young's modulus and high stretchability under room temperature (Fig. 2c). The Young's modulus of the film is calculated to be 43.68 ± 3.27 MPa (mean ± s.d., $n = 5$) from the low strain region (<10% strain) of the stress–strain curve at a stretching speed of 50 mm min$^{-1}$, indicating the high binding strength of the metal–ligand interaction. The toughness is calculated by integrating the area of the strain–stress curve under the stretching speed of 50 mm min$^{-1}$, which is as high as 29.3 MJ m$^{-3}$. Such a high toughness is among the highest reported[23]. The stress–strain curves consist of an initial stiffening region (where the tension

significantly increases with the increase of the strain), followed by a yield point, then a dependence on the strain rates. When the strain speed increases, less time is allowed for the ligand exchange processes and re-formation of the complexes, which reduce the fracture tolerance and increase the tensile stress. At high strain rates (higher than 50 mm min$^{-1}$), the stress decreased firstly and then increased until reaching the point of breaking. At low strain rates (lower than 25 mm min$^{-1}$), however, the stress continuously decreased until rupture. The high speed dependence of stress–strain relation can be attributed to the dynamic exchange process between the ligand and Zn(II), which is in good agreement with the rheological results. The stretchability of the **Zn(Hbimcp)$_2$-PDMS** film is also dependent on the stretching speed. Upon decreasing the stretching speed from 100 to 10 mm min$^{-1}$, the stretchability increased but the stress at break decreased. Such stretching speed dependence is typical for polymer films with dynamic interactions[12,26]. A maximum stretchability more than 2400% can be achieved for a sample of 0.8 mm thickness, 5 mm gauge length, and 5 mm width at a stretching speed of 10 mm min$^{-1}$ (Fig. 2d). Moreover, a polymer strip with dimensions of 30 mm (L) × 5 mm (W) × 1 mm (H) can sustain a load of 1000 g (Fig. 2e).

Our **Zn(Hbimcp)$_2$-PDMS** film not only has a high stretchability and high toughness, but also exhibits self-healing capability at room temperature. To demonstrate the self-healing capability, the polymer film was cut into two pieces and subsequently put together to allow healing at different time and then applied the strain–stress measurement under the stretching speed of 50 mm min$^{-1}$ (Fig. 2f). The cut on the film was observed to almost disappear after healing at room temperature for 24 h, although minor scars were still visible. In order to make the cut region more distinguishable, one of the two pieces was stained using a blue dye (Supplementary Figure 13). As shown in Supplementary Figure 14, the healed film can again sustain a large strain after a 24 h healing duration at room temperature. As expected, a longer healing time resulted in higher recovered fracture strain. Healing at room temperature for 24 h led to a recovered fracture strain of (1060 ± 80)% and a high healing efficiency ($\eta$)[19] of (98.9 ± 1.9)% (mean ± s.d., $n = 5$, Fig. 2f, Supplementary Table 1). Moreover, two undamaged polymer films can be joined together through self-healing, suggesting the existence of dynamic exchanging of metal–ligand coordination bonds in the polymer matrix (Supplementary Figure 15). This autonomous self-healing polymer with such a high stretchability and toughness is a rare combination (Supplementary Table 2).

**Mechanism study**. The excellent mechanical robustness and self-healing properties of **Zn(Hbimcp)$_2$-PDMS** film should be owing to the stable but dynamic bonding features of the [Zn(Hbimcp)$_2$]$^{2+}$ complex. The association constant of [Zn(Pr-Hbimcp)$_2$]$^+$ was measured by UV–Vis spectroscopy, which shows a relatively large value of 2.2 × 10$^{11}$ (Supplementary Figure 16 and Supplementary Figure 17, see the Supplementary Information for details). The Zn-Hbimcp complex may offer multiple mechanisms for energy dissipation: (i) stereochemical structure transformation through the diagonal twist of the complex (Fig. 3a) and (ii) dynamic rupture and reconstruction of the Zn-Hbimcp bonding configurations during chain unfolding and sliding (Fig. 3b). These features make this material able to extend to a large length and give rise to its high toughness. When damaged upon stretching or cutting, the Zn-Hbimcp bond can quickly reform through intermolecular ligand exchange. Therefore, the material can be self-healed at room temperature. The Zn-Hbimcp coordination bonds are diverging, so that the breakage, reformation, and exchange of the bonds can take place more readily.

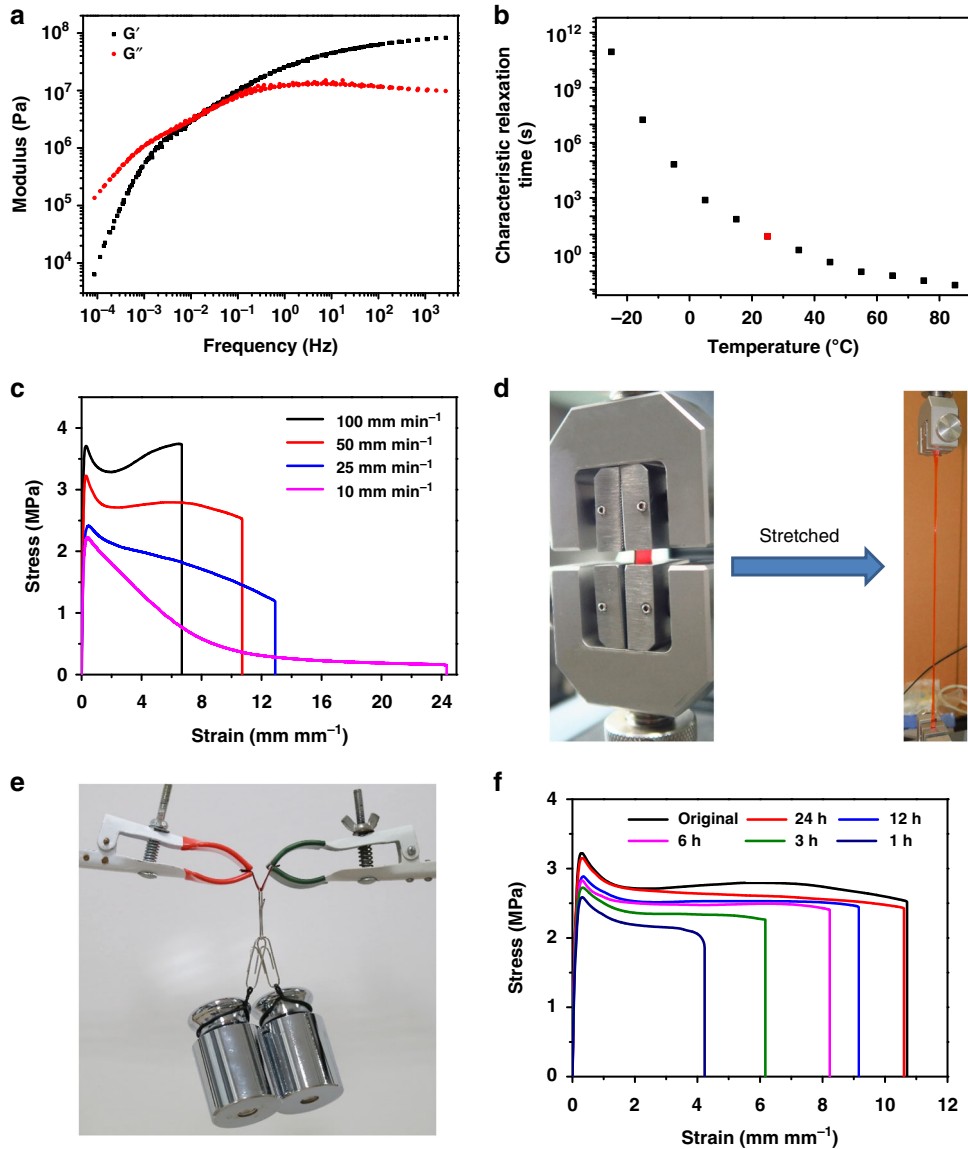

**Fig. 2** Mechanical properties of Zn(Hbimcp)$_2$-PDMS polymer. **a** The time–temperature superposition curve of **Zn(Hbimcp)$_2$-PDMS** polymer at 25 °C. **b** The characteristic relaxation times of **Zn(Hbimcp)$_2$-PDMS** polymer under different temperature. **c** The strain–stress test of **Zn(Hbimcp)$_2$-PDMS** polymer under different stretching speed. **d** Photographs of a film before and after stretching. **e** Optical image of a film sustaining a 1000 g load. **f** Strain–stress curves of a film healed at room temperature (25 °C) for different lengths of time show an increase of the stretching ability when the film is allowed to heal for a longer time

We performed several control experiments in order to validate our proposed Zn-Hbimcp bond reformation/switching mechanism. Firstly, we used a metal-to-ligand molar ratio of 1:1 in the preparation. According to the characterization on model complexes (see Supplementary Note 1), the di-nuclear complex [Zn$_2$(bimcp)$_2$Cl$_2$] was the dominant species in this condition. In [Zn$_2$(bimcp)$_2$Cl$_2$], all the binding sites in the alterdentate ligands are coordinated to Zn(II) ions. The association constant was measured to be $3.7 \times 10^{17}$. The di-nuclear complex is diphenoxo-bridged and therefore no intramolecular ligand exchange can take place (Fig. 3c, Supplementary Figure 3b). Unlike in [Zn(Hbimcp)$_2$]$^{2+}$ in which the intramolecular ligand exchange can accelerate the intermolecular ligand exchange due to the generation of meta-stable three-coordinated intermediates (Supplementary Figure 18), the intermolecular ligand exchange in [Zn$_2$(bimcp)$_2$Cl$_2$] is slow and has to be activated by heating or solvation effect. Next, we used Ni(II) instead of Zn(II) for our

polymer (Fig. 3d). It is known that Ni(II) forms planar complexes with Hbimcp ligands[33]. The planar Ni(II) planar complexes have no stereochemical isomer due to steric hindrance. Therefore, energy dissipation through stereochemical structure transformation is unfavored for **Ni(Hbimcp)$_2$-PDMS** polymer. The association constant of the model complex [Ni(Pr-Hbimcp)$_2$]$^{2+}$ was also measured by the UV–Vis spectroscopy, which shows a relatively larger value of $5.4 \times 10^{11}$ (Supplementary Figure 19). All these control metal–ligand complexes gave higher modulus but lower fracture strain (**Zn$_2$(bimcp)$_2$-PDMS** was broken at 350% strain with stress-at-break of 5.8 MPa, while **Ni(Hbimcp)$_2$-PDMS** was broken at 20% strain with stress-at-break of 4.2 MPa) polymers, which show inferior self-healing ability at room temperature, manifesting the importance of the unique Zn-Hbimcp dynamic coordination features (Fig. 3e). Besides, the **Zn$_2$(bimcp)$_2$-PDMS** polymer shows much lower healing efficiency (<2% at 25 °C, Supplementary Figure 20), much longer

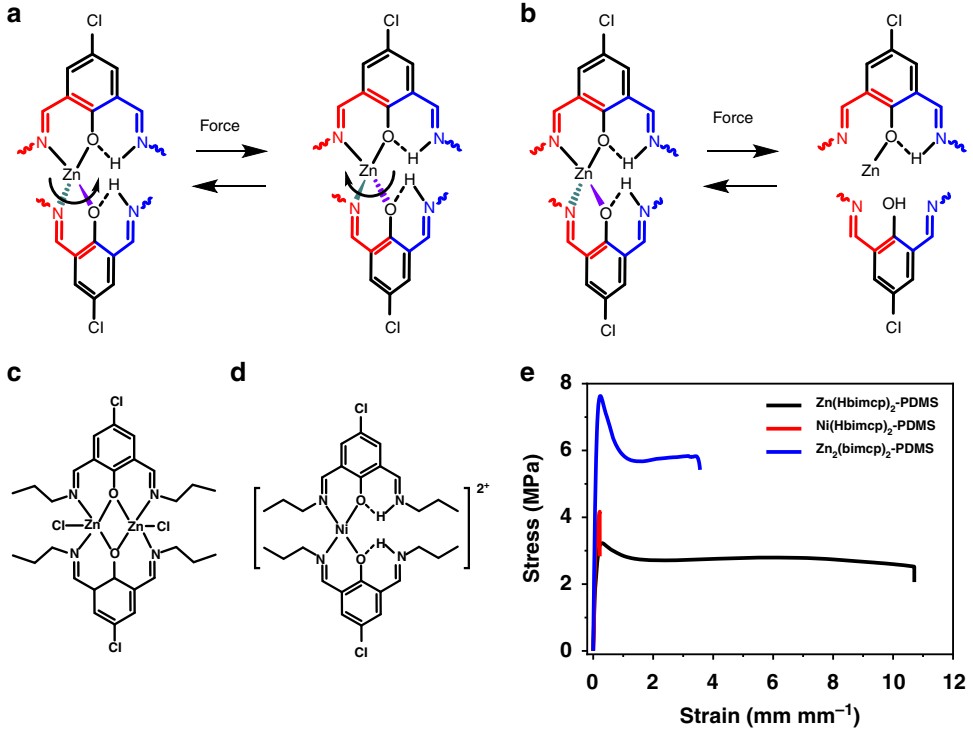

**Fig. 3** Control experiments for mechanism study. **a, b** Two energy dissipation process for [Zn(Hbimcp)$_2$]$^{2+}$. **c** The molecular structure of [Zn$_2$(bimcp)$_2$Cl$_2$]. **d** The molecular structure of [Ni(Hbimcp)$_2$]$^{2+}$. **e** Comparison of stress–strain curves for **Zn(Hbimcp)$_2$-PDMS**, **Ni(Hbimcp)$_2$-PDMS** polymer, and **Zn$_2$(bimcp)$_2$-PDMS**

characteristic relaxation time ($4 \times 10^4$ s at 25 °C, Supplementary Figure 21), and larger residual stress than **Zn(Hbimcp)$_2$-PDMS** (at 25 °C, Supplementary Figure 22). The mechanical properties of the polymer with the different metal ion to ligand molar ratio show that increasing the number of Zn(II) centers makes the films stronger and stiffer, while decreasing the number of Zn(II) centers makes the films softer and more stretchable (Supplementary Figure 23).

**Energy absorbing properties**. Energy absorbing materials are of great interest in numerous applications, such as protecting the drivers and passengers in car crashes when used as glass coating, preventing impact injuries in dangerous sports, reducing shock for delicate instruments, and safeguarding soldiers from projectiles and blasting[39–41]. Viscoelastic polymers are excellent candidates for energy absorbing materials due to their light weight, mechanical flexibility, and high energy absorbing capacity. The key feature of this kind of energy absorbing materials is energy dissipation through viscous deformation and bond breakage. The **Zn(Hbimcp)$_2$-PDMS** polymer is tough and viscoelastic (will not restore its configuration after stretching, Supplementary Figure 24) and can therefore make an excellent candidate for energy absorbing material. Cyclic stress–strain tests with a maximum applied strain from 25 to 500% showed pronounced hysteresis, indicating energy dissipation due to bond exchange (Fig. 4a and Supplementary Figure 25). The cyclic stress–strain curves with multiple cycles at different strain rate show that the hysteresis depends on the loading rate and loading cycles (Supplementary Figure 26 and Supplementary Figure 27). The energy absorbing efficiency $\omega$ can be calculated from the integrated area of loading strain–stress curves $W_1$ and unloading strain–stress curves $W_2$ as Eq. 1:

$$\omega = \frac{W_1 - W_2}{W_1} = \frac{dW}{W_1} \tag{1}$$

The $\omega$ was calculated to be 90%, such a high energy absorbing efficiency was scarce in the literature[42]. Stress–relaxation experiments showed that the samples undergo substantial stress relaxation with time. The residual stress rapidly decreased to lower than 5% after 45 s, 95 s, 165 s, and 205 s with 25, 50, 100, and 200% strain, respectively (Fig. 4b). Creep-recovery experiments of **Zn(Hbimcp)$_2$-PDMS** polymer also showed a high residual strain (more than 80%) that was not recovered within the experimental time (Supplementary Figure 28). All these features indicate that this polymer is promising for energy absorbing application.

It is known that both elastic and viscoelastic materials have their own advantages and disadvantages when being used as energy absorbing material. Elastic materials will deform under load and return to their original state once the load is removed, but the energy absorbing capacity is limited as there is no energy dissipation process. Viscoelastic materials can absorb and dissipate energy efficiently through bond breakage and viscous deformation, but they do not recover their shape when the load is removed. To overcome this problem, we coated a layer of **Zn(Hbimcp)$_2$-PDMS** polymer on elastic polyurethane sponge by a "dip and dry" method. The **Zn(Hbimcp)$_2$-PDMS** polymer has an excellent film-forming ability (Supplementary Figure 29) and therefore it formed a thin-layer coating on the polyurethane surface while keeping the porous structure of polyurethane sponge (Supplementary Figure 30), and the weight ratio of the coated polymer could be controlled by the times of dip and dry process. Figure 4c shows that the energy absorption density of the 100 wt% **Zn(Hbimcp)$_2$-PDMS**-coated sponge (composite sponge) is about 11.14 mJ cm$^{-3}$, which is 4 times the blank sponge (2.86 mJ cm$^{-3}$). The energy absorption density of the coated sponge could be tuned by adjusting the thickness of coated **Zn(Hbimcp)$_2$-PDMS** (Supplementary Figure 31 and Supplementary Figure 32). Interestingly, the deformed composite sponge can

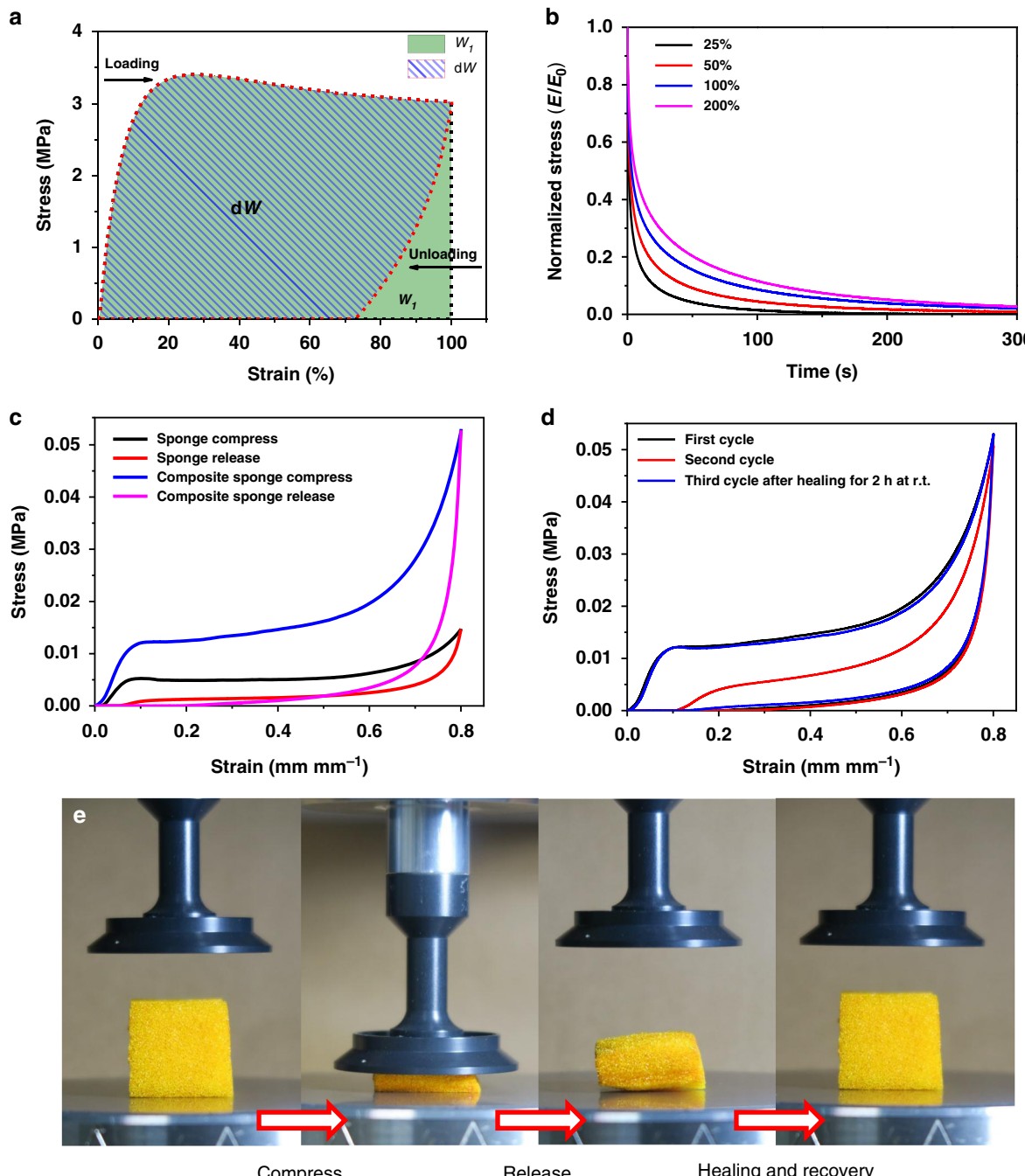

**Fig. 4** The energy absorbing properties of Zn(Hbimcp)$_2$-PDMS polymer. **a** The calculation of energy absorbing efficiency. **b** Normalized stress–relaxation curves under different strain. **c** Cyclic strain–stress curves of blank sponge and composite sponge. **d** Cyclic strain–stress curves of composite sponge with healing process. **e** Optical images of the composite sponge under compression, healing, and recovery processes

recover its shape and mechanical properties within 2 h (Fig. 4d, e, Supplementary Figure 33-35 and Supplementary Movie 1). Therefore, the advantages of both elastic and viscoelastic materials are combined in the composite sponge. Such energy absorbing materials can be very useful for car crash protection, sportswear, shockproof pad, armored clothing, etc.

## Discussion
In order to address the conundrum between mechanical and self-healing properties of a material, we designed and synthesized a PDMS polymer containing thermodynamically stable while kinetically labile coordination complex, Zn(II)-2,6-bis((propyli-mino)methyl)-4-chlorophenol (denoted as Zn(Pr-bimcp)$_2$). The Zn(Pr-bimcp)$_2$ has a relatively large association constant (2.2 × 10$^{11}$) but also exhibits fast and reversible intra-/inter-molecular ligand exchange process. The as-prepared **Zn(Hbimcp)$_2$-PDMS** polymer is highly stretchable (can be stretched to 2400% strain), tough (with the toughness of 29.3 MJ m$^{-3}$), and can be autono-mously self-healed at room temperature. Control experiments, by varying the metal ions and metal-to-ligand molar ratios, showed that the bond strength and bond dynamics play the key role in determining the material's mechanical and self-healing property. The design concept presented provides an approach to

self-healing polymers with excellent mechanical properties, while the polymer reported in this work can find promising applications as anti-impact materials.

## Methods

**Materials and general measurements.** Poly(dimethylsiloxane) bis(3-aminopropyl) terminated ($H_2N$-PDMS-$NH_2$, $M_n = 700$–900) was purchased from Gelest. Toluene was distilled from Na before use. The remaining commercially available reagents were used as received (Sigma-Aldrich). [1]H NMR spectra (400 MHz) were recorded on a Bruker DRX 400 NMR spectrometer in deuterated solvents at 25 °C. The chemical shift data for each signal are reported in units of $\delta$ (ppm) relative to tetramethylsilane as an internal standard. FTIR spectra were recorded on a Bruker Tensor27 FTIR spectrophotometer by 64 scans from 4000 to 600 $cm^{-1}$, with a resolution of 4 $cm^{-1}$. ESI-MS were recorded with the methanol as solvent and mobile phase. The measurements were carried out under the positive ion mode with the ionization temperature of 290 °C. Gel permeation chromatography (GPC) was measured with a Malvern Viscotek GPCmax/VE2001 connected to a Triple detection array (TDA 305). Differential scanning calorimetry (DSC) curves were measured with a Mettler DSC 1 analyzer under a dry nitrogen atmosphere (50 ml $min^{-1}$). The experimental temperature is between −100 °C and 120 °C with a ramp rate of 20 °C $min^{-1}$ under a nitrogen atmosphere. The measurements of each sample were performed with two cycles and the curves of the second cycle were used. Creep-recovery experiments were performed on rectangular samples (ca. 1 mm (T) × 4 mm (W) × 8 mm (L)) using TA-Q800 DMA. The samples were equilibrated for 5 min at a specified temperature, then pulled in constant stress and held. After 30 min, the stress was released and the samples were allowed to relax for an additional 60 min. The dynamic compressive tests (drop impact tests) were performed on square samples (ca. 20 mm (T) × 100 mm (W) × 100 mm (L)) with the drop height of 100 mm and drop weight of 7.2 kg using Instron Dynatup 9250 HV at 25 °C. The cyclic compressive tests were performed using Instron 3343 at 25 °C with the square samples (ca. 20 mm (T) × 20 mm (W) × 20 mm (L)) at the different strain rate of 5, 10, 25, 50, 100, 200, and 400 mm $min^{-1}$, respectively. The duration time at full compression was 0, 1, 5, and 60 s, respectively. The rheological properties were studied by a DHR-2 Rheometer (TA Instruments). Uniaxial tensile measurements and stress-relaxation analysis were performed on an Instron 3343 equipped with a 500 N load cell with a specific strain rate at 25 °C. Single-molecule force experiments on **Hbimcp-PDMS** and **Zn(Hbimcp)₂-PDMS** macromolecules were performed on a modified atomic force microscopy as reported previously[12]. The crystal structures were determined at 296 K on a Bruker SMART CCD diffractometer using monochromated Mo K$\alpha$ radiation ($\lambda = 0.71073$ Å). The cell parameters were retrieved using SMART software and refined using *SAINT* for all observed reflections. The photographs and videos were taken by digital cameras. More details of general measurements are given in the Supplementary Note 2.

**Synthesis of 5-chloro-2-hydroxyisophthalaldehyde.** The 5-chloro-2-hydroxyisophthalaldehyde was synthesized according to ref. [43] with a small modification. 4-Chlorophenol (12.8 g, 100 mmol) and hexamethylenetetramine (28.0 g, 200 mmol) were dissolved in anhydrous trifluoroacetic acid (200 ml) under argon atmosphere, the resulting solution was refluxed for 24 h, and the color changed from yellow to reddish-orange. The mixture was poured into 4 M HCl (600 ml) and stirred for 5 min, after which the solution was settled for 24 h and yellow crystal precipitated out. The precipitates were collected by filtration and washed with deionized water and hexane. The product was purified by recrystallized in ethyl alcohol. The resulting yellow crystals were filtered off and dried in vacuum oven at 70 °C to give pure 5-chloro-2-hydroxyisophthalaldehyde. Yield: 8.3 g (45%); [1]H NMR (400 MHz, CDCl₃, δ): 11.58 (s, 1 H), 10.20 (s, 2 H), 8.01 (s, 2 H); FTIR (KBr): $\nu = 1776$ (s), 1738 (m), 1375 (s), 1354 (s), 875 (m), 759 (m), 684 $cm^{-1}$ (s) (Supplementary Figure 36a and Supplementary Figure 37).

**Synthesis of 2,6-bis((propylimino)methyl)-4-chlorophenol (Pr-Hbimcp).** 5-Chloro-2-hydroxyisophthalaldehyde (1.84 g, 10 mmol) and propylamine (1.18 g, 20 mmol) were dissolved in anhydrous ethanol (40 ml) under stirring, and then acetic acid (200 μl) was added into the solution as a catalyst. The resulting mixture was stirred for 4 h at 65 °C under argon atmosphere. After the reaction, the solution was concentrated by rotary evaporation at 45 °C and the reddish-orange oil was obtained. The products were purified by column chromatography with ethyl acetate: hexane = 1:5 as eluent. The final product was obtained as reddish-orange oil. Yield: 2.3 g (86%). [1]H NMR (400 MHz, CDCl₃, δ): 14.89 (s, 1 H), 8.60 (s, 2 H), 7.86 (s, 2 H), 3.56 (t, 4 H), 1.64 (m, 4 H), 0.91 (t, 6 H) (Supplementary Figure 36b and Supplementary Figure 38).

**Synthesis of 2,6-bis((benzylimino)methyl)-4-chlorophenol (Bz-Hbimcp).** This compound was prepared similarly to Pr-Hbimcp, starting from 5-chloro-2-hydroxyisophthalaldehyde (1.84 g, 10 mmol), phenylmethanamine (2.14 g, 20 mmol), and acetic acid (200 μl) in anhydrous ethanol (40 ml), but utilized a different workup procedure. After stirring for 4 h at 65 °C under argon atmosphere, the mixture was cooled down and yellow crystal precipitated out. The precipitates were collected by filtration and washed with a small amount of ethanol for 3 times.

The product was purified by recrystallized in ethyl alcohol. The resulting yellow crystals were filtered off and dried in vacuum oven at 70 °C to give pure Bz-Hbimcp. Yield: 3.0 g (83%); [1]H NMR (400 MHz, CDCl₃, δ): 14.59 (s, 1 H), 8.76 (s, 2 H), 7.76 (s, 2 H), 7.36 (m, 8 H), 7.30 (m, 2 H),4.82 (s, 4 H) (Supplementary Figure 36c and Supplementary Figure 39).

**Synthesis of Hbimcp-PDMS ligand.** The three-neck round-bottomed flask was equipped with a Dean-Stark trap, an argon inlet, and a reflux condenser. 5-Chloro-2-hydroxyisophthalaldehyde (1.84 g, 10 mmol) dissolved in anhydrous toluene (20 ml) was added to a solution of $H_2N$-PDMS-$NH_2$ (14 g, $M_n = 700$–900) in anhydrous toluene under stirring, and then catalyst acetic acid (200 μl) was added into the solution. The resulting mixture was stirred for 12 h at 130 °C under argon atmosphere. After the reaction, the toluene was evaporated by rotary evaporation at 60 °C, then 20 ml $CH_2Cl_2$ was added to dissolve the product, and 60 ml MeOH was poured into the reactor to separate the product. Red precipitate-like viscous liquid appeared and the mixture was settled for 30 min. The upper clear solution was then decanted and the purified product was obtained. The dissolution–precipitation–decantation process was repeated for three times and the final product was subjected to vacuum evaporation to remove the solvent and trace of acetic acid. Yield 11.9 g (76%). [1]H NMR (400 MHz, CDCl₃, δ): 14.35 (s, 1 H), 8.36 (s, 2 H), 7.48 (s, 2 H), 3.49 (t, 4 H), 1.64 (m, 4 H), 0.49 (t, 4 H). Molecular weight according to GPC: $M_w = 21,000$; $M_n = 15,000$ ($Đ = 1.4$) (Supplementary Figure 36d, Supplementary Figure 40 and Supplementary Figure 41).

**Synthesis of small molecule model compounds and polymer complexes.** The typical procedure to prepare the model compounds is: a certain amount of the ligand (Pr-Hbimcp or Bz-Hbimcp, 1 mmol) was dissolved in 10 ml dichloromethane (DCM), proportioning anhydrous $ZnCl_2$ was dissolved in the methanol with the concentration of 0.25 g $ml^{-1}$ and added dropwise to the ligand solution. The resulting solution was stirred for 2 h. After the reaction, the solvent was removed under reduced pressure. The residue was washed with the diethyl ether and filtered. The as-prepared solid was used for further characterization and crystallization.

The typical procedure to prepare the complex polymers is: a certain amount of the anhydrous $ZnCl_2$ (0.25 g $ml^{-1}$) solution in methanol (determined by the molar ratio of Hbimcp ligand to Zn(II)) was added to a solution of polymer ligand **Hbimcp-PDMS** (1 g) in DCM (10 ml). The mixed solution was stirred for 12 h at room temperature and then concentrated to about 2 ml. The concentrated solution was poured into a polytetrafluoroethene (PTFE) mold measuring 36 mm length × 20 mm width × 3.0 mm height and dried at room temperature for 1 day followed by drying at 70 °C for 12 h. The as-prepared polymer film has a size of 36 mm length × 20 mm width × 1.0 mm height. The polymer films were then peeled off from the PTFE mold for further testing.

**Fabrication of composite sponge.** A dip and dry process was used to fabricate the composite sponge. Firstly, 10 g of **Zn(Hbimcp)₂-PDMS** polymer was dissolved in 200 ml of dichloromethane solvent, and the blank polyurethane sponge was dipped into the solution. After 1 min, the sponge was taken out and centrifuged under the speed of 1000 rpm to remove the residual solution filled in the hole of sponge. Then, the sponge was dried under vacuum to remove the dichloromethane and get the **Zn(Hbimcp)₂-PDMS**-coated sponge. The dip and dry process could be repeated for several times to control the weight percent of **Zn(Hbimcp)₂-PDMS** in the composite sponge.

## Data availability

The data that support the findings of this study are available from the corresponding author upon reasonable request. The crystallographic data in this study have been deposited at the Cambridge Crystallographic Data Centre (CCDC), under deposition number 1886156 for Zn(Pr-Hbimcp)Cl₂, 1817755 for Zn₂(Pr-Hbimcp)₂Cl₂, and 1886157 for [Zn₂(Pr-bimcp)(Pr-Hbimcp)₂(CH₃O)](ClO₄)₂. These data can be obtained free of charge from the Cambridge Crystallographic Data Centre via www.ccdc.cam.ac.uk/data_request/cif.

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

## Acknowledgements

This work was supported by the National Natural Science Foundation of China (Grant No. 21631006 and 21771100), the Natural Science Foundation of Jiangsu Province (Grant No. BK20170016 and BK20151377), the project of the Scientific and Technological Support Program in Jiangsu Province (Grant No. BE2014147-2), the Fundamental Research Funds for the Central Universities (020514380147), and the National Postdoctoral Program for Innovative Talents (BX20180136). We thank Professor Xiao-Liang Wang for valuable discussion on the rheological measurement.

## Author contributions

J.-C.L., C.-H.L., J.-L.Z. and Z.B. conceived, designed, and directed the project; J.-C.L., X.-Y.J., D.-P.W. and Y.-B.D. performed the experiments; J.-C.L., X.-Y.J., D.-P.W., Y.-B.D., P.Z., C.-H.L., J.-L.Z. and Z.B. analyzed the data; J.-C.L., C.-H.L., J.-L.Z. and Z.B. wrote the paper. All authors discussed the results and commented on the manuscript.
