## [Peer Review File · Nature Communications]

Reviewers' comments:

Reviewer #1 (Remarks to the Author):

The manuscript "Thermodynamically stable..." by Lai et al reports the method to overcome the trade-off between the mechanical properties and dynamic nature of self-healing by the design and synthesis of a polymer containing thermodynamically stable whilst kinetically labile coordination complex of Zn-based, three-dimensional, alterdentate ligands. The authors focus on the tunable nature of unique noncovalent interactions between a metal ion and its surrounding organic ligands. Several material designs based on metal-ligand coordination using different metal ions and ratios are selected and they are introduced as crosslinking units into PDMS polymer to synthesize self-healing polymers. For those material designs and synthesized polymers, various spectroscopic analyses such as NMR, ESI-MS, FT-IR, UV-vis and force spectroscopy are well conducted. Also, the mechanical and rheological properties are well-studied. Its scientific novelty is well-recognized. Furthermore, the authors suggest their potential applications as energy absorbing materials together with elastic polyurethane sponge. Although the work contains no apparent serious flaw, several critical issue and minor issues as listed below should be addressed before the publication of this manuscript in Nature Communications.

1. The authors measured association constant to evaluate thermodynamic properties. However, the authors didn't attempt to characterize kinetic properties except for attaching references related to alterdentate ligands.

Is it possible to quantitatively measure and compare (alterdentate) ligand exchange constant (or comparable study)? The reviewer wishes the authors to find suitable experiments. (How about compare time/temperature-dependent NMR results of the mixtures of $Zn1-(Pr-bimcp)_2/Zn1-(Bz-bimcp)_2$ vs. $Zn2-(Pr-bimcp)_2/Zn2-(Bz-bimcp)_2$? The exchange reaction rate should be faster in case of $Zn1-(Pr-bimcp)_2/Zn1-(Bz-bimcp)_2$ rather than $Zn2-(Pr-bimcp)_2/Zn2-(Bz-bimcp)_2$.)

2. The authors are highly encouraged to conduct more experiments using Zn-bimcp-PDMS coated sponge as energy absorbing materials to emphasize its high novelty and impact. For example, drop a heavy stuff onto the both Zn-bimcp-PDMS coated sponge and reference sponge, and compare their shock absorbing capabilities. The data in current form is not enough to make the potential readers impressed.

- More about energy absorbing experiment, the reviewer wants to check the effect of compression speed (or the duration time at full compression). Does the polymer coated sponge return to its original shape directly when the speed is faster (or duration time is shorter)? Or more lately return to original shape when the speed is lower (or duration time is longer)?

- Please describe the experimental procedures in detail in the Supporting Information (including compression speed, duration time at full compression).

3. If alterdentate concept designed by the authors is working, the self-healing properties of $Zn2-bimcp_2$ -PDMS ($Zn:bimcp$ -PDMS=10:10) should be inferior. The tensile-experiments of $Zn2-bimcp_2$ -PDMS like Fig 2f and Fig 4b can be a supportive data.

4. The reviewer wants to know the reason why the authors designed a polymer structure containing (5-chloro-2-oxy-1,3-phenylene)dimethanimine, especially, N-O-N chelation.

5. The reviewer found a similar reference (Nat. Commun. 2018, 9, 2725) by Prof. Jing-Lin Zuo. It is recommended to make mention of this ref. in the main text; about differences of association constant, mechanical, self-healing properties. The reviewer is concerned why Zn-bimcp polymer (this study) exhibited lower tensile strength, though it has a higher association constant than PDMS-COO-Zn (ref.).

6. The accessibility for references of 34,31,32 is low. How about describing "alterdentate ligand" concept in Supporting Information?

Minorities.

1. In abstract, it is recommended not to use abbreviations such as "Zn-bimcp".
2. Fig. 1d: missing chlorine.
3. In conclusion, mis-spelling "toughness", not "roughness".
4. Please check chemical formula $[Zn_2(bimcp)_2Cl_2]^{2+}$ (should not be $[Zn_2(Pr-bimcp)_2Cl_2]$).
5. Ref [28] 2012 -> 2002
6. Supporting Figure 18. Check strain unit (% -> mm/mm)
7. The denotation in the legend of Figure 3e should be modified from "Zn²⁺-bimcp₂-PDMS" to "Zn₂(bimcp)₂-PDMS".
8. The definition of energy absorbing efficiency η , based on W_1 and dW , must be given in the main text rather than in the supplementary information. And above energy absorbing efficiency η should be separated with healing efficiency η (Line 228).
9. In the Supplementary Figure 19, the applied stress of red-colored curve is expressed as 250 kPa in the legend while it is expressed as 200 kPa in the caption. The authors should correct this.

Reviewer #2 (Remarks to the Author):

The present manuscript by Z. Bao et al. describes the synthesis of self-healing polymers based on PDMS and ligands – namely bimcp, which is capable of complexing zinc ions. The synthesis of these novel materials is described in detail. Additionally, the complexation of the metal ions was investigated revealing the structure of the complexes. The mechanical properties of the polymers as well as the ability for self-healing was studied extensively. Noteworthy, the resulting materials feature very good mechanical properties, in particular a very high the toughness. Scratch healing as well as healing of specimen was investigated. Furthermore, these materials feature very good energy absorbing properties, which was finally demonstrated by the coating of a sponge. Overall the study is sound and leaves only a few minor open questions open (see below).

Considering the present literature the novelty / originality of the present study is somewhat questionable. The general design principle of the present materials is in line with prior examples published by the authors. PDMS is chain extended by a ligand motif and the resulting materials are complexes with the appropriate metal ions:

- Nature Chemistry volume 8, pages 618–624 (2016): $[Fe(Hpdca)_2]^+$
- Inorg. Chem. 2018, 57, 3232–3242: DMS-NNN/PDMS-MeNNN with zinc, which feature also a imine moiety
- Chem. Commun., 2015, 51, 8928: cobalt(II) triazole complexes
- Macromol. Rapid Commun. 2016, 37, 952–956: iron triazole complexes
- J. Am. Chem. Soc. 2016, 138, 6020–6027: bpy with zinc and iron
- JPSA 2017, 55, 3110–3116: bpy with Eu(III) and Tb(III).

If this submission would be for NPG Asia Mater. the present version would be fine. However, for Nature Commun. a more detailed comparison revealing the benefits (toughest material, ...) of the present study in comparison to the prior examples is required.

Minor issues:

- The authors depict the complexes as the charged species with protons. Alternatively, the complexes could also be neutral (zinc (II) and the phenolates. What is the evidence for these charged complexes? If it is positively charged, what are the counter ions?
- Considering the proton at the imine (see above) imine metathesis might also contribute to the healing process in the solid state. Imines, in particular in acidic media, are prone to hydrolysis or can self-exchange. Have the authors observed an exchange? Is the molar mass distribution of the polymer changing? (decomplexation by HEDTA and SEC measurement of the resulting linear polymer)
- Force spectroscopy measurements have been performed. Have the authors compared these results with the contour length of the polymers? For PDMS with higher molar masses around 12

nm are given in literature (DOI: 10.1038/NCHEM.2492). With the smaller molar mass in the present study much smaller values could be expected. However, this effect seems not to be present in the measured curves, which feature much larger distances.

- The rupture force of the zinc complexes could be compared with literature values of other complexes. Are these values high or low?
- The healing efficiencies for the other healing times could be given in the Supporting Information.
- The higher zinc content in Zn₂(bimcp)₂-PDMS makes the polymer films stiffer compared to the Zn(bimcp)₂-PDMS. How would the Zn_{1.5}(bimcp)₂-PDMS behave? A smaller amount of Zn₂(bimcp)₂-complexes could potentially reinforce the polymer.

Supporting Information

- Is the dispersity of the PDMS known?
- Figure 1: Where do the protons come from? At low temperature there are two protons, the imines are depicted as protonated form. These moieties would be charged positively. These protons could / should also appear in the NMR spectrum.
- ¹³C NMR data could be added
- Figure 10: A scale bar for the microscopic images is missing
- Figure 11: The strain could be added
- Figure 16: The legend (top left) is missing
- General comment for the NMR spectra: Sometimes the integral (green line) seems to be smaller than the corresponding peaks (consequently lowering the numbers) – for instance Figure 25 peak at around 1 ppm.

Reviewer #3 (Remarks to the Author):

The authors report a design principle to synthesize a strong, stretchable, and selfhealable elastomer using Zn-bimcp coordination bonds. This kind of coordination complex is thermodynamically stable whilst kinetically labile. They use PDMS elastomer as a representative material and show combined excellent mechanical properties and selfhealing capabilities. The PDMS elastomer exhibits Young's modulus~43.68 MPa, stretchability~24, toughness~29.3 MJ/m³, and self-healing efficiency~98.9%. This type of material is potentially used as energy absorbing material in many fields.

The reviewer suggests the paper be published if the authors can address the following issues.

- In the introduction part, the authors are recommended to give additional explanation on why the Zn-bimcp coordinate system is unique, such as high association energy and high healing efficiency, etc., compared to other metal-coordination systems, disulfide bonds and many types of hydrogen bonds that are described in the reference 13, 18, 23, and 24.
- In Fig. 1e, two peak forces are observed in stretching bimcp-PDMS polymer chain, what is the event corresponding to each peak?
- In Fig. S7, why is the force in second stretch much lower than the first stretch? Are those continuous tests in a same polymer chain?
- In describing Fig. 2C, please add explanation on how the Zn-bimcp plays role in the rate-dependent stress-strain curves?
- In Fig.2C, the Zn-bimcp-PDMS demonstrates excellent stretchability, even with 2400% strain at a loading rate of 10 mm/min, does it restore its initial configuration given enough time? Does the hysteresis also depend on the loading rate? How does the hysteresis change with loading cycles? The reviewer suggests conducting stress-strain curves with multiple cycles for each strain rate.

- In the self-healing experiment, do the authors press the damaged parts together to make them contact intimately? Are there gaps between the healing interface that hinder healing?
- In the coating of sponge, what is the typical thickness of the coating? How does the quality fraction of coated Zn-bimcp-PDMS influence the coating thickness?
- On line 195, "module" should be "modulus", please carefully check typos in the manuscript.

Point-by-point response to reviewers' comments

Reviewer #1 (Remarks to the Author):

The manuscript “Thermodynamically stable...,” by Lai et al reports the method to overcome the trade-off between the mechanical properties and dynamic nature of self-healing by the design and synthesis of a polymer containing thermodynamically stable whilst kinetically labile coordination complex of Zn-based, three-dimensional, alterdentate ligands. The authors focus on the tunable nature of unique noncovalent interactions between a metal ion and its surrounding organic ligands. Several material designs based on metal-ligand coordination using different metal ions and ratios are selected and they are introduced as crosslinking units into PDMS polymer to synthesize self-healing polymers. For those material designs and synthesized polymers, various spectroscopic analyses such as NMR, ESI-MS, FT-IR, UV-vis and force spectroscopy are well conducted. Also, the mechanical and rheological properties are well-studied. Its scientific novelty is well-recognized. Furthermore, the authors suggest their potential applications as energy absorbing materials together with elastic polyurethane sponge. Although the work contains no apparent serious flaw, several critical issue and minor issues as listed below should be addressed before the publication of this manuscript in Nature Communications.

Response: We appreciate your positive comments.

1. The authors measured association constant to evaluate thermodynamic properties. However, the authors didn't attempt to characterize kinetic properties except for attaching references related to alterdentate ligands.

Is it possible to quantitatively measure and compare (alterdentate) ligand exchange constant (or comparable study)? The reviewer wishes the authors to find suitable experiments. (How about compare time/temperature-dependent NMR results of the mixtures of $Zn1-(Pr-bimcp)_2/Zn1-(Bz-bimcp)_2$ vs. $Zn2-(Pr-bimcp)_2/Zn2-(Bz-bimcp)_2$? The exchange reaction rate should be faster in

case of Zn1-(Pr-bimcp)2/Zn1-(Bz-bimcp)2 rather than Zn2-(Pr-bimcp)2/Zn2-(Bz-bimcp)2.)

Response:

Thank you for your insightful comments. We agree that it would be very helpful to characterize the kinetic properties of the model complexes. We have tried time- and temperature-dependent ^1H NMR to quantitatively measure the ligand exchange constant but got no useful results because we can not obtain the mono-nuclear complexes $\text{Zn}_1(\text{Hbimcp})_2$ and di-nuclear complex $\text{Zn}_2(\text{bimcp})_2$ as pure products. No matter what metal-to-ligand molar ratios we used, we got $\text{Zn}_1(\text{Hbimcp})_2$, $\text{Zn}_2(\text{bimcp})_2$, $\text{Zn}_2(\text{bimcp})_3$ and other unknown products as a mixture, with different content percentages when different amount of ZnCl_2 was added (Figure R1 and R2). The structure diversity of Zn- bis(imino)phenol has been frequently reported in the literature (*Inorg. Chem.* **2005**, 44, 147-157; *Inorg. Chem.* **2008**, 47, 11711-11719; *Dalton Trans.* **2012**, 41, 1889–1896). We have tried to separate and characterize these species for a very long time but failed. Therefore, the ^1H NMR of $\text{Zn}_2(\text{Hbimcp})_2$ is not available. The $[\text{Zn}(\text{Pr-Hbimcp})_2]^{2+}$ complex, which we believed is pure and used it for variable temperature ^1H NMR in our original manuscript, turns out to be $\text{Zn}(\text{Pr-Hbimcp})\text{Cl}_2$ with a metal-to-ligand molar ratio of 1:1. Actually, the crystals of $\text{Zn}(\text{Hbimcp})\text{Cl}_2$ were obtained in most cases no matter what metal-to-ligand molar ratio we used. The crystal structure of $\text{Zn}(\text{Pr-Hbimcp})\text{Cl}_2$ was determined by single crystal X-ray crystallography. Although we got the crystal structure of $\text{Zn}_2(\text{Pr-bimcp})_2\text{Cl}_2$ and $[\text{Zn}_2(\text{Pr-bimcp})(\text{Pr-Hbimcp})_2(\text{CH}_3\text{O})](\text{ClO}_4)_2$ accidentally in our experiment, which indicate the structure diversity in Zn-Hbimcp coordination system, these complexes can not be separated as pure product in large amount. $\text{Zn}(\text{Pr-Hbimcp})\text{Cl}_2$ is the only clean and characterizable product both in solution and at solid state. The purity can be evidenced by comparison of the X-ray powder diffraction peaks of the experimental and simulated data (Figure R3).

Figure R1. Molecular and crystal structures of (a) $\text{Zn}(\text{Pr-Hbimcp})\text{Cl}_2$, (b) $\text{Zn}_2(\text{Pr-bimcp})_2\text{Cl}_2$ and (c) $[\text{Zn}_2(\text{Pr-bimcp})(\text{Pr-Hbimcp})_2(\text{CH}_3\text{O})](\text{ClO}_4)_2$.

Figure R2. ESI-MS spectra of the reaction product between Zn(II) complexes and Pr-Hbimcp ligand with different molar ratio. a, 2:1; b, 3:2; c, 1:1; d, the possible structure of the obtained peaks. The inserted figure is the partial enlargement of the main figure. The asterisks represent the peak that cannot be attributed at present. The intense peaks at 267.2 belong to the free ligand generated due to electrospray ionization.

Figure R3. Comparison of the X-ray powder diffraction peaks of the experimental and simulated data of $\text{Zn}(\text{Pr-Hbimcp})\text{Cl}_2$.

However, we believe the lack of time- and temperature-dependent ^1H NMR of $\text{Zn}_1(\text{Hbimcp})_2$ and $\text{Zn}_2(\text{bimcp})_2$ will not affect our conclusion in our manuscript. The mono-nuclear complex $\text{Zn}(\text{Pr-Hbimcp})\text{Cl}_2$ showed rapid ligand exchange reaction above room temperature, indicating that the $\text{Zn}(\text{II})\text{-Hbimcp}$ complexes are highly dynamic. As for the ligand exchange reaction rate, we can get a quantitative evaluation by comparing the relaxation time from stress-relaxation test and time-temperature superposition data of the polymers. As shown in Figure R4a, the stress of $\text{Zn}_2(\text{bimcp})_2\text{-PDMS}$ polymer decrease much slower than that of $\text{Zn}(\text{Hbimcp})_2\text{-PDMS}$ at different initial strain. The residual stress is still significant after 3600 s (Figure R4b), which indicated there was no obvious ligand exchange phenomenon in the polymer of $\text{Zn}_2(\text{bimcp})_2\text{-PDMS}$ within 1 h. Temperature dependent characteristic relaxation time can also manifest the different ligand exchange speed in these polymers. As derived from time-temperature superposition (TTS) of rheological data, the characteristic relaxation time of $\text{Zn}_2(\text{bimcp})_2\text{-PDMS}$ at 25 °C is as long as 4×10^4 s (Figure R5), which is much longer than $\text{Zn}(\text{Hbimcp})_2\text{-PDMS}$ (8 s).

Figure R4. The stress-relaxation curves of Zn₂(bimcp)₂-PDMS at different initial strain. a, compare with the curves of Zn(Hbimcp)₂-PDMS with the duration time of 300 s. b, with the duration time of 3600s.

Figure R5. The time-temperature superposition curve of Zn₂(bimcp)₂-PDMS polymer at 25 °C.

Based on these observations, we believe that there are two dynamic ligand exchange processes: intra-molecular ligand exchange and inter-molecular ligand exchange. For Zn(Hbimcp)Cl₂ and [Zn(Hbimcp)₂]²⁺, both intra-molecular ligand exchange and inter-molecular ligand exchange are possible (Figure R6). But for [Zn₂(bimcp)₂]²⁺, only the inter-molecular ligand exchange is possible since all the

binding sites in the alterdentate ligands are occupied (Figure R7). The intra-molecular ligand exchange is fast and can be observed at room temperature. Moreover, the intra-molecular ligand exchange can accelerate the inter-molecular ligand exchange due to the generation of meta-stable three-coordinated intermediates (Figure R6). In contrast, the inter-molecular ligand exchange is slow and has to be activated by heating or solvation effect. That is why the $Zn_2(\text{bimcp})_2\text{-PDMS}$ polymer show much longer relaxation time. We have added these data and discussions in the revised manuscript.

Figure R6. The intra-molecular and inter-molecular ligand exchange process in $Zn\text{-Hbimcp}$ complexes.

Figure R7. The inter-molecular ligand exchange process in $Zn_2(bimcp)_2$ complexes.

2. The authors are highly encouraged to conduct more experiments using Zn-bimcp-PDMS coated sponge as energy absorbing materials to emphasize its high novelty and impact. For example, drop a heavy stuff onto the both Zn-bimcp-PDMS coated sponge and reference sponge, and compare their shock absorbing capabilities. The data in current form is not enough to make the potential readers impressed.

- More about energy absorbing experiment, the reviewer wants to check the effect of compression speed (or the duration time at full compression). Does the polymer coated sponge return to its original shape directly when the speed is faster (or duration time is shorter)? Or more lately return to original shape when the speed is lower (or duration time is longer)?

- Please describe the experimental procedures in detail in the Supporting Information (including compression speed, duration time at full compression).

Response:

Thank you for your helpful suggestion. We have conducted additional experiments to characterize the energy absorbing properties of $Zn(Hbimcp)_2$ -PDMS coated sponge and made supplements in the revised manuscript:

- 1) According to the dynamic compression tests (drop impact tests), with increasing the weight ratio of $\text{Zn}(\text{Hbimcp})_2\text{-PDMS}$, the composite sponges showed higher strain of impact but lower stress of impact (Figure R8), indicating that coating $\text{Zn}(\text{Hbimcp})_2\text{-PDMS}$ onto sponges decreased the damage resistance but increased the damage tolerance. This observation is reasonable because the $\text{Zn}(\text{Hbimcp})_2\text{-PDMS}$ polymer has weaker strength but better energy absorbing capacity as compared to polyurethane polymer. Such performance changes are useful in some applications, such as for helmet liners, where damage tolerance is more important than damage resistance.

Figure R8. Strain-stress curves of composite sponge with different weight fraction of coated $\text{Zn}(\text{Hbimcp})_2\text{-PDMS}$ under the dynamic compressing test.

- 2) We have checked the effects of compression speed and duration time at full compression on the energy absorbing properties of the $\text{Zn}(\text{Hbimcp})_2\text{-PDMS}$ coated sponge (100 wt%). The results showed that the residual strain and the configuration recovery time of $\text{Zn}(\text{Hbimcp})_2\text{-PDMS}$ coated sponge decreased when the compression speed was higher or the duration time was shorter (Figure R9a and Figure R10a), but there is no obvious effect on the blank sponge (Figure R9b and Figure R10b). This observation is also reasonable because less time is allowed for the ligand exchange processes and re-formation of the complexes at

higher compression speed or shorter duration time, thus the viscoelastic Zn(Hbimcp)₂-PDMS behaves like an elastic polymer. Therefore, there is no significant residual strain and the configuration of the Zn(Hbimcp)₂-PDMS coated sponge can be recovered quickly.

Figure R9. Cyclic strain-stress curves of the Zn(Hbimcp)₂-PDMS coated sponge (a) and blank sponge (b) with different strain rate.

Figure R10. Cyclic strain-stress curves of the Zn(Hbimcp)₂-PDMS coated sponge (a) and blank sponge (b) with duration time at full compression.

3) The experimental procedures (including drop height, compression speed and duration time at full compression) are now described in detail in the Supporting Information.

3. If alterdentate concept designed by the authors is working, the self-healing properties of Zn₂-bimcp₂-PDMS (Zn:bimcp-PDMS=10:10) should be inferior. The tensile-experiments of Zn₂-bimcp₂-PDMS like Fig 2f and Fig 4b can be a supportive data.

Response:

According to your suggestion, we have investigated the self-healing and stress-relaxation properties of Zn₂(bimcp)₂-PDMS. The self-healing properties of Zn₂(bimcp)₂-PDMS were indeed inferior. We have partially damaged the Zn₂(bimcp)₂-PDMS polymer (remain 10% uncut to keep the incision better contact) and measured the self-healing properties at room temperature (25 °C). As shown in Figure R11, the healing efficiency for 48 h was still very low (< 2%). The stress-relaxation results showed that the stress of Zn₂(bimcp)₂-PDMS polymer decreased much slower than that of Zn(Hbimcp)₂-PDMS at different initial strains. The residual stress was still significant after 3600 s (Figure R4b), which indicated there was no obvious ligand exchange phenomenon in the polymer of Zn₂(bimcp)₂-PDMS within 1 h. As derived from time-temperature superposition (TTS) of rheological data, the characteristic relaxation time of Zn₂(bimcp)₂-PDMS at 25 °C is as long as 4×10^4 s (Figure R5), which is much longer than Zn(Hbimcp)₂-PDMS (8 s). These data indicate that although the inter-molecular ligand exchange process may be possible in the polymer film, the speed is too slow. That is why the Zn₂(bimcp)₂-PDMS polymer showed inferior self-healing properties.

Figure R11. The self-healing properties of $Zn_2(\text{bimcp})_2\text{-PDMS}$ at 25 °C. **b** is the partial enlargement of **a**.

4. The reviewer wants to know the reason why the authors designed a polymer structure containing (5-chloro-2-oxy-1,3-phenylene)dimethanimine, especially, N-O-N chelation.

Response:

From our previous study, we found that the mechanical properties of a polymer are determined by the thermodynamic stability of the crosslinking sites. The more stable (i.e. higher association constant) of the crosslinking sites, the stronger and tougher but less dynamic of the polymer. In contrast, the self-healing rate of a polymer is determined by the kinetic lability of the crosslinking sites. Therefore, to achieve both high toughness/high modulus while having a rapid self-healing rate, a molecular design concept for the crosslinking site that is both thermodynamically stable whilst kinetically labile is needed. Alterdentate ligand can provide two equivalent donor centers. However, due to the steric hindrance, the two equivalent donor centers can not coordinate with the same metal ion at the same time and therefore the two coordination atoms are alternative and interchangeable. Take N-O-N chelation for example, when one imine-N is coordinated with Zn, the other uncoordinated imine-N has the ability to replace the previous coordinated imine-N and coordinate with Zn. We envisage that such a unique coordination system would be an ideal crosslinking site that is both thermodynamically stable and kinetically labile, and can lead to strong and tough self-healing polymers. That's why we designed the polymer structure containing (5-chloro-2-oxy-1,3-phenylene)dimethanimine ligands. We have revised the introduction part of the manuscript to make this design concept more clear to the readers.

5. The reviewer found a similar reference (Nat. Commun. 2018, 9, 2725) by Prof. Jing-Lin Zuo. It is recommended to make mention of this ref. in the main text; about differences of association constant, mechanical, self-healing properties. The reviewer

is concerned why Zn-bimecp polymer (this study) exhibited lower tensile strength, though it has a higher association constant than PDMS-COO-Zn (ref.).

Response:

The mechanical properties of a polymer are determined not only on the strength but also on the density of the crosslinking sites. In this study, although the crosslinking site has a higher association constant as compared to PDMS-COO-Zn (*Nat. Commun.* **2018**, 9, 2725), the density of the crosslinking site is much less. Therefore, the degree of freedom of the polymer segment between the two crosslinking sites is higher, leading to higher tensile strain and higher toughness. The description of the reference (*Nat. Commun.* **2018**, 9, 2725) has added in the main text as “Recently, we reported a rigid and healable polymer, which cross-linked by weak (with association constant of $4.10 \times 10^4 \text{ M}^{-1}$) but abundant Zn(II)-carboxylate interactions. However, this polymer shows low fracture strain and can not be healed at room temperature.”

6. The accessibility for references of 34,31,32 is low. How about describing “alterdentate ligand” concept in Supporting Information?

Response:

Thank you for pointing this out. Although the accessibility for references of 34,31,32 is low, the concept of “alterdentate ligand” have been well documented in Ref. 35 by Prof. Von Zelewsky. According to your suggestion, we have added the description of “alterdentate ligand” in the Supporting Information as “According to the reference 35, the definition of “alterdentate ligand” is that a species which offers to a metal ion more than one equivalent coordination site. In an “alterdentate ligand” there is, principally, always a re-arrangement possible in which the metal is transferred from one site to another one. This can be either an inter- or intra- molecular process. The rearrangement reaction is kinetically controlled by the activation energy and entropy experienced by the metal on the reaction path. The free energy difference is zero by definition, if the coordination sites are equivalent.”

Minorities.

1. In abstract, it is recommended not to use abbreviations such as “Zn-bimcp”.

Response: Thank you for pointing out this mistake. We have added the full name of the abbreviation in the abstract.

2. Fig. 1d: missing chlorine.

Response: Thank you for pointing out this mistake. We have added the label for chlorine on the structure.

3. In conclusion, mis-spelling “toughness”, not “roughness”.

Response: Thank you for pointing out this mistake. We have replaced “roughness” with “toughness” in the revised manuscript.

4. Please check chemical formula $[\text{Zn}_2(\text{bimcp})_2\text{Cl}_2]^{2+}$ (should not be $[\text{Zn}_2(\text{Pr-bimcp})_2\text{Cl}_2]$).

Response: We have checked the crystal structure of $[\text{Zn}_2(\text{Pr-bimcp})_2\text{Cl}_2]$, and the chemical formula should be $[\text{Zn}_2(\text{Pr-bimcp})_2\text{Cl}_2]$ because the hydrogen of phenolic hydroxyl group in Pr-Hbimcp have been replaced by the Zn(II) when the complex was formed.

5. Ref [28] 2012 -> 2002

Response: Thank you for pointing out this mistake. The mistake has been corrected.

6. Supporting Figure 18. Check strain unit (% -> mm/mm)

Response: Thank you for pointing out this mistake. The strain unit has been modified to mm/mm.

7. The denotation in the legend of Figure 3e should be modified from “Zn₂-bimcp₂-PDMS” to “Zn₂(bimcp)₂-PDMS”.

Response: Thank you for pointing out this mistake. The “Zn₂-bimcp₂-PDMS” has been modified as “Zn₂(bimcp)₂-PDMS”.

8. The definition of energy absorbing efficiency η , based on $W1$ and dW , must be given in the main text rather than in the supplementary information. And above energy absorbing efficiency η should be separated with healing efficiency η (Line 228).

Response: Thank you for your suggestion. We have given the definition of energy absorbing efficiency ω in the main text and change the symbol for energy absorbing efficiency as ω .

9. In the Supplementary Figure 19, the applied stress of red-colored curve is expressed as 250 kPa in the legend while it is expressed as 200 kPa in the caption. The authors should correct this.

Response: Thank you for pointing out this mistake. The have corrected this number.

Reviewer #2 (Remarks to the Author):

The present manuscript by Z. Bao et al. describes the synthesis of self-healing polymers based on PDMS and ligands – namely bimcp, which is capable of complexing zinc ions. The synthesis of these novel materials is described in detail. Additionally, the complexation of the metal ions was investigated revealing the structure of the complexes. The mechanical properties of the polymers as well as the ability for self-healing was studied extensively. Noteworthy, the resulting materials feature very good mechanical properties, in particular a very high the toughness. Scratch healing as well as healing of specimen was investigated. Futhermore, these materials feature very good energy absorbing properties, which was finally demonstrated by the coating of a sponge. Overall the study is sound and leaves only a few minor open questions open (see below).

Considering the present literature the novelty / originality of the present study is somewhat questionable. The general design principle of the present materials is in line with prior examples published by the authors. PDMS is chain extended by a ligand

motif and the resulting materials are complexes with the appropriate metal ions:

- Nature Chemistry volume 8, pages 618–624 (2016): [Fe(Hpdca)₂]⁺
- Inorg. Chem. 2018, 57, 3232–3242: DMS-NNN/PDMS-MeNNN with zinc, which feature also a imine moiety
- Chem. Commun., 2015, 51, 8928: cobalt(II) triazole complexes
- Macromol. Rapid Commun. 2016, 37, 952–956: iron triazole complexes
- J. Am. Chem. Soc. 2016, 138, 6020–6027: bpy with zinc and iron
- JPSA 2017, 55, 3110–3116: bpy with Eu(III) and Tb(III).

If this submission would be for NPG Asia Mater. the present version would be fine. However, for Nature Commun. a more detailed comparison revealing the benefits (toughest material, ...) of the present study in comparison to the prior examples is required.

Response: Thank you for your positive comments and helpful suggestions. Yes, there are already several literatures reporting self-healing PDMS polymers crosslinked by coordination bonds. However, as shown in Table R1, the comparison between this manuscript and the prior published results revealed that the polymer reported in this work has the highest Young's modulus, toughness, healing efficiency and competitive healing temperature, healing time, maximum strain. The excellent mechanical and self-healing properties was achieved through the design of a thermodynamically stable whilst kinetically labile coordination system, which has never been reported before.

Table R1. The comparison between this manuscript and the prior published results

Ref. No.	ligand	metal salt	Young's modulus (MPa)	Maximum strain (mm mm ⁻¹)	Healing temperature (°C)	Healing time (h)	Healing efficiency (%)	toughness (MJ m ⁻³)
this work	Hbimecp	ZnCl ₂	43.68 ± 3.27	24	25 (R.T.)	24	98.9 ± 1.9	29.3
1	pdca	FeCl ₃	0.54 ± 0.1	45 ± 0.2	25 (R.T.)	48	90.0±3.0	3.96 ^a
2	NNN	ZnCl ₂	0.12 ± 0.01	2.3 ± 0.26	25 (R.T.)	12	91.3 ± 2.2	0.15± 0.01
	Me-NNN	ZnCl ₂	0.067 ± 0.012	4.56 ± 0.33	25 (R.T.)	0.5	94.7 ± 2.8	0.23±0.03
3	triazole	CoCl ₂	1.12	5.7	120	24	47.3	3.17 ^a
4	triazole	FeCl ₃	0.46 ± 0.1	34	60	20	94.3	6.81 ^a
5	bpy	FeCl ₂	0.9± 0.2	1.25±0.20	25 (R.T.)	48	9 ^a	0.52 ^a
	bpy	Fe(BF ₄) ₂	1.0± 0.15	1.10 ± 0.16	25 (R.T.)	48	12 ^a	0.18 ^a
	bpy	ZnCl ₂	1.2± 0.21	1.43±0.2	25 (R.T.)	48	21 ± 3	0.57 ^a
	bpy	Zn(ClO ₄) ₂	1.2± 0.15	2.95 ± 0.17	25 (R.T.)	48	55 ± 21	1.98 ^a
	bpy	Zn(OTf) ₂	1.1± 0.2	3.1 ± 0.15	25 (R.T.)	48	76 ± 22	2.53 ^a
6	bpy	Eu(NO ₃) ₃	1.2 ± 0.14	1.05± 0.2	25 (R.T.)	24	35 ^a	0.62 ^a
	bpy	Eu(OTf) ₃	1.0± 0.26	3.00±0.1	25 (R.T.)	24	94 ^a	0.54 ^a
	bpy	Tb(NO ₃) ₃	1.2± 0.18	1.20±0.20	25 (R.T.)	24	32 ^a	0.54 ^a
	bpy	Tb(OTf) ₃	1.0± 0.08	2.90±0.15	25 (R.T.)	24	96 ^a	0.57 ^a

Notes: ^aThis data was estimated according to the data reported in the reference.

References: 1) Li C. H., *et al.* A highly stretchable autonomous self-healing elastomer. *Nat. Chem.* **8**, 618-624 (2016); 2) Wang D. P., Lai J. C., Lai H. Y., Mo S. R., Zeng K. Y., Li C. H., Zuo J. L. Distinct Mechanical and Self-Healing Properties in Two Polydimethylsiloxane Coordination Polymers with Fine-Tuned Bond Strength. *Inorg. Chem.* **57**, 3232–3242 (2018); 3) Jia X. Y., Mei J. F., Lai J. C., Li C. H., You X. Z. A self-healing PDMS polymer with solvatochromic properties. *Chem. Commun.* 51, 8928-8930 (2015); 4) Jia X. Y., Mei J. F., Lai J. C., Li C. H., You X. Z. A Highly Stretchable Polymer that Can Be Thermally Healed at Mild Temperature. *Macromol. Rapid Comm.* **37**, 952-956 (2016); 5) Rao Y.L., *et al.* Stretchable Self-Healing Polymeric Dielectrics Cross-Linked Through Metal-Ligand Coordination. *J. Am. Chem. Soc.* **138**, 6020-6027 (2016); 6) Rao Y. L., Feig V., Gu X., Wang G.J., Bao Z. The effects of counter anions on the dynamic mechanical response in polymer networks crosslinked by metal–ligand coordination. *J. Polym. Sci., Part A: Polym. Chem.* **55**, 3110-3116 (2017).

Minor issues:

- The authors depict the complexes as the charged species with protons. Alternatively, the complexes could also be neutral (zinc (II) and the phenolates). What is the evidence for these charged complexes? If it is positively charged, what are the counter ions?

Response:

Thank you for your insightful comments. We have performed additional measurements in order to determine the accurate structure of the complexes.

For $[\text{Zn}(\text{Hbimcp})]\text{Cl}_2$ type complexes, we have measured the ^1H NMR of $[\text{Zn}(\text{Pr-Hbimcp})]\text{Cl}_2$ (Figure R12) and $[\text{Zn}(\text{Bz-Hbimcp})]\text{Cl}_2$ (Figure R13) again. This time we use DMSO as solvent to observe the signals of active hydrogen, which are hard to observe in protic solvents. As show in Figure R12 and Figure R13, the signals of active hydrogen at 13.46 and 14.32 illustrated that no neutralization reactions occurred when the complexes formed, indicating that the Zn(II) complexes is charged. The counter ion of the formed complexes is Cl^- , which comes from the reactant of ZnCl_2 . The crystal structure of $[\text{Zn}(\text{Pr-Hbimcp})]\text{Cl}_2$ complexes given in the Figure R1 could also give the evidence. Such structure can be supported by literature studies where the phenol group was not deprotonated when base (such as triethylamine) was not added in the solution (*Aust. J. Chem.* **2003**, 56, 703–706).

Figure R12. ^1H NMR for $[\text{Zn}(\text{Pr-Hbimcp})]\text{Cl}_2$ with the solvent of DMSO.

Figure R13. ^1H NMR for $[\text{Zn}(\text{Bz-Hbimcp})]\text{Cl}_2$ with the solvent of DMSO.

For $[\text{Zn}_2(\text{bimcp})_2]\text{Cl}_2$ type complexes, although we were not able to get the ^1H NMR spectra to prove the absence of signals of active hydrogen, the crystal structure clearly shows that the phenol group was deprotonated. This is due to that, when both the O and N atoms are coordinated to Zn(II) metal ions, the O-H or N-H bonds are significantly weakened and the H atom is easy to leave although no base was added. Such structure can also be supported by literature (*Inorg. Chem.* **2002**, 41, 6426-6431).

Based on these observations, we can conclude that for Zn-Hbimcp complexes with unbounded N atoms, the phenol group was not deprotonated. The H atom was located between O and N atoms (more close to N atoms as revealed by single crystal X-ray crystallography). For Zn-Hbimcp complexes without unbounded N atoms, the phenol group was not protonated. In order to differentiate these structures, we denote Zn-Hbimcp complexes with protonated phenol group as Zn(Hbimcp), while the Zn-Hbimcp complexes with deprotonated phenol group as Zn(bimcp) (Figure R14)

Figure R14. The structure of Zn(Hbimcp) and Zn(bimcp).

- Considering the proton at the imine (see above) imine metathesis might also contribute to the healing process in the solid state. Imines, in particular in acidic media, are prone to hydrolysis or can self-exchange. Have the authors observed an exchange? Is the molar mass distribution of the polymer changing? (decomplexation by HEDTA and SEC measurement of the resulting linear polymer)

Response:

Thank you for your helpful suggestion. In order to determine if imines hydrolysis or self-exchange contribute to the healing process of our polymer, we have conducted additional experiments to characterize the hydrolysis and self-exchange properties of the small molecule and polymer containing the Hbimcp ligand:

- 1) We mixed the solution of small molecule Pr-Hbimcp and Bz-Hbimcp at room temperature for 4 h and then conducted the ^1H NMR test with the solvent of DMSO. The spectrum of the mixed solution was shown in Figure R15. The new peaks at 14.64, 8.76, 8.58, and 4.79 manifest the presence of self-exchange process, indicating that the phenolic hydroxyl group on the ligand of Hbimcp could provide the acidic environment to catalyze the reaction. However, there was no new peak at about 10.2 ppm (the characteristic peak of the aldehyde group), indicating that there is no hydrolysis in the solution;

Figure R15. ^1H NMR for Pr-Hbimcp (**a**), Bz-Hbimcp (**b**) and the mixed solution (**c**) with the solvent of DMSO. **d** is the partial enlargement of **c**.

2) We tracked the changes of the molar mass distribution of the polymer Hbimcp-PDMS by GPC under different condition. One is the as prepared sample,

one is the sample stored at room temperature for 7 days, and the other one is the sample complexation by ZnCl_2 and then decomplexation by terpyridine in the dichloromethane solution (HEDTA was not adopted because it is insoluble in dichloromethane). The results showed that there were no obvious molar mass distribution changes under the different conditions (Figure R16), indicating that the self-exchange process in the polymer is unobvious.

Figure R16. GPC elution curves of Hbimcp-PDMS under different conditions.

Based on the above observations, we believe that imine metathesis do not contribute significantly to the healing process for $\text{Zn}(\text{Hbimcp})_2\text{-PDMS}$ polymer. Presumably due to that the PDMS matrix is not favorable for imine exchange since the imine exchange rate is sensitive to the environment (*Macromolecules* **2016**, 49, 6277–6284). Such conclusion can be evidenced by other observations. First, as we stated in our original manuscript, the polymer $\text{Zn}_2(\text{bimcp})_2\text{-PDMS}$ and $\text{Ni}(\text{Hbimcp})\text{-PDMS}$ do not show self-healing behavior although they contain the same Hbimcp ligand with imine groups. Moreover, we crosslinked the bis(3-aminopropyl) terminated linear oligomer $\text{H}_2\text{N-Hbimcp-PDMS-NH}_2$ with tri-functional homopolymer of hexamethylene diisocyanate (THDI) to obtain the crosslinked polymer THDI-Hbimcp-PDMS and measured the self-healing properties at room

temperature (Figure R17). We also did not observe obvious self-healing phenomenon even after two days (Figure R18).

Figure R17. Synthesis of the crosslinked polymer THDI-Hbimcp-PDMS.

Figure R18. The mechanical and self-healing properties of THDI-Hbimcp-PDMS with tensile speed of 50 mm min^{-1} and healing temperature of $25 \text{ }^\circ\text{C}$ for different time.

- Force spectroscopy measurements have been performed. Have the authors compared these results with the contour length of the polymers? For PDMS with higher molar masses around 12 nm are given in literature (DOI: 10.1038/NCHEM.2492). With the smaller molar mass in the present study much smaller values could be expected. However, this effect seems not to be present in the measured curves, which feature much larger distances.

Response: Thank you for the constructive comments. The contour length increments (the increased lengths caused by the unfolding of the polymer chain, denoted as ΔL_c) was calculated from the force–extension curves (Figure 1e) and found to be 14, 66, and 78 nm, respectively (Figure R19). This contour length increments are quite similar to those for Fe-Hpdca-PDMS (*Nat. Chem.* **2016**, 8, 618-624) although the molecular lengths of the repeating units ($\text{NH}_2\text{-PDMS-NH}_2$) are different (about 2 nm for $\text{Zn(Hbimcp)}_2\text{-PDMS}$ and 12 nm for Fe-Hpdca-PDMS). This is due to that, with the shorter repeating units, the polymer chain in $\text{Zn(Hbimcp)}_2\text{-PDMS}$ become quite rigid. Only the Hbimcp ligands separated by several $\text{NH}_2\text{-PDMS-NH}_2$ repeating units can be connected through coordination to Zn(II) metal ions in the same polymer chain. (Figure R20). Therefore, the contour length increments for $\text{Zn(Hbimcp)}_2\text{-PDMS}$ will not show significant difference with those of Fe-Hpdca-PDMS.

Figure R19. The contour length increment (ΔL_c) of $\text{Zn}(\text{Hbimcp})_2$ -PDMS polymer derived from single-molecule (single chain) force spectroscopy measurement

Figure R20. The different folding modes for Fe-Hpdca-PDMS and $\text{Zn}(\text{Hbimcp})_2$ -PDMS polymers.

- The rupture force of the zinc complexes could be compared with literature values of other complexes. Are these values high or low?

Response: The rupture force of the zinc complexes is about 108.5 ± 40.9 pN, which is similar to the Fe(III)-pdca complexes in our previous study (103 ± 12 pN, *Nat. Chem.* **2016**, 8, 618-624), but lower than the ferric-thiolate bonds (160 ± 60 pN at pH = 6 and 211 ± 86 pN at pH = 7.4, *Nat. Commun.* **2015**, 6:7569) and gold-thiolate bonds (165 ± 55 pN, *J. Am. Chem. Soc.* **2015**, 137, 15358–15361). However, it should be noted that the rupture force from AFM study reveals the kinetics of the transition from the bound state to the unbound state over a potential energy surface. The rupture forces will vary significantly with different loading rates and probe stiffnesses (*Methods* **2013**, 60, 142–150). Therefore, the force measured in different situations and for different systems can not be directly compared. What we can know for sure from these data is that the single-chain of Fe-Hpdca-PDMS and Zn(Hbimcp)₂-PDMS molecule can be unfolded and quickly refolded due to the dynamic rupture and reconstruction of coordination bonds.

- The healing efficiencies for the other healing times could be given in the Supporting Information.

Response: Thank you for your suggestions. The healing efficiencies for the other healing times have been given in the Supporting Information and list in Table R2.

Table R2. Healing efficiencies of Zn(Hbimcp)₂-PDMS with various healing time.

Healing time	Maximal strength (MPa)	Breaking strength (MPa)	Breaking strain (mm mm ⁻¹)	Healing efficiency (%)
Original	3.22	2.52	10.71	-
24h	3.15 ± 0.34	2.42 ± 0.27	10.60 ± 0.80	98.9 ± 1.9
12h	2.88 ± 0.26	2.44 ± 0.21	9.17 ± 0.67	85.6 ± 2.4
6h	2.82 ± 0.29	2.40 ± 0.35	8.24 ± 0.73	76.9 ± 3.7
3h	2.72 ± 0.32	2.26 ± 0.31	6.17 ± 0.58	57.6 ± 1.3
1h	2.58 ± 0.23	1.89 ± 0.29	4.23 ± 0.74	39.5 ± 2.1

- The higher zinc content in $Zn_2(\text{bimcp})_2$ -PDMS makes the polymer films stiffer compared to the $Zn(\text{bimcp})_2$ -PDMS. How would the $Zn_{1.5}(\text{bimcp})_2$ -PDMS behave? A smaller amount of $Zn_2(\text{bimcp})_2$ -complexes could potentially reinforce the polymer.

Response: Yes, a smaller amount of $Zn_2(\text{bimcp})_2$ -complexes could reinforce the polymer films and make them stiffer. As shown in Supplementary Figure 21, with increasing the content of Zn(II), the obtained polymers exhibited the higher modulus and maximal strength, but the breaking strains were also declined.

Supporting Information

- Is the dispersity of the PDMS known?

Response: The dispersity index of the starting polymer H_2N -PDMS- NH_2 is 1.3 as provided by the supplier. The dispersity index of the synthetic polymer Hbimcp-PDMS is about 1.4, which was calculated from the GPC elution curve in Supplementary Figure 39.

- Figure 1: Where do the protons come from? At low temperature there are two protons, the imines are depicted as protonated form. These moieties would be charged positively. These protons could / should also appear in the NMR spectrum.

Response: The protons were from the phenolic hydroxyl group after coordination. The signals of protons appeared in the 1H NMR spectra with the solvent of DMSO, which showed in Figure R12 and R13.

- ^{13}C NMR data could be added

Response: The ^{13}C NMR data has been added in the Supplementary information.

- Figure 10: A scale bar for the microscopic images is missing

Response: The scale bar for the microscopic has been added in Supplementary Figure 10 (Supplementary Figure 13 in the revised manuscript).

- Figure 11: The strain could be added

Response: The strain has been added in supplementary Figure 11 (Supplementary Figure 14 in the revised manuscript).

- Figure 16: The legend (top left) is missing.

Response: The legend (top left) has been added in Supplementary Figure 16 (Supplementary Figure 19 in the revised manuscript).

- General comment for the NMR spectra: Sometimes the integral (green line) seems to be smaller than the corresponding peaks (consequently lowering the numbers) – for instance Figure 25 peak at around 1 ppm.

Response: Thank you for your kind remind. The integral have been adjust to fit each peaks.

Reviewer #3 (Remarks to the Author):

The authors report a design principle to synthesize a strong, stretchable, and self-healable elastomer using Zn-bimcp coordination bonds. This kind of coordination complex is thermodynamically stable whilst kinetically labile. They use PDMS elastomer as a representative material and show combined excellent mechanical properties and self-healing capabilities. The PDMS elastomer exhibits Young's modulus~43.68 MPa, stretchability~24, toughness~29.3 MJ/m³, and self-healing efficiency~98.9%. This type of material is potentially used as energy absorbing material in many fields. The reviewer suggests the paper be published if the authors can address the following issues.

Response: Thank you for your positive comments.

- In the introduction part, the authors are recommended to give additional explanation on why the Zn-bimcp coordinate system is unique, such as high association energy and high healing efficiency, etc., compared to other metal-coordination systems,

disulfide bonds and many types of hydrogen bonds that are described in the reference 13, 18, 23, and 24.

Response: Thank you for your comments. We have given additional explanation on why the Zn-Hbimcp coordinate system is unique in the introduction part as “To demonstrate the concept, we designed and synthesized a poly(dimethylsiloxane) (PDMS) polymer containing an alterdentate ligand, 2,6-bis((propylimino)methyl)-4-chlorophenol (denoted as Hbimcp). This ligand can provide two equivalent imine-N donor centers. However, due to the steric hindrance, the two equivalent imine-N donor centers can not coordinate with the same metal ion (Zn(II) in our study) at the same time and therefore the two coordination atoms are alternative and interchangeable. On the other hand, the Zn(II)-2,6-bis((propylimino)methyl)-4-chlorophenol (denoted as Zn(Pr-Hbimcp)₂) complex has a relatively large association constant. Such a unique coordination system would be an ideal crosslinking site that is both thermodynamically stable and kinetically labile, and can lead to strong and tough self-healing polymers which can not be easily achieved through designing of covalent bonds (such as disulfide bonds) or non-covalents interactions (such as hydrogen bonds).”

- In Fig. 1e, two peak forces are observed in stretching bimcp-PDMS polymer chain, what is the event corresponding to each peak?

Response: The first peak can be assigned to the force when the AFM cantilever tip was in contact with the substrate. The second peak can be assigned to the force when the polymer chain was detached from the glass substrate.

- In Fig. S7, why is the force in second stretch much lower than the first stretch? Are those continuous tests in a same polymer chain?

Response: In Fig. S7 (Supplementary Figure 10 in the revised manuscript), we didn't put the two curves in the same vertical axis but use the same scale bar in order to make the two curves distinguishable to readers. The force in the second stretch is

actually similar to the first stretch. The continuous tests were performed with the same polymer chain.

- In describing Fig. 2C, please add explanation on how the Zn-bimcp plays role in the rate-dependent stress-strain curves?

Response:

Thank you for your suggestion. When the strain speed increases, as manifested by the curve at 100 mm min^{-1} , less time is allowed for the ligand exchange processes and re-formation of the complexes, which reduce the fracture tolerance and increase the tensile stress. In contrast, when the strain speed decreases, as manifested by the curve at 10 mm min^{-1} , more time is allowed for the ligand exchange processes and re-formation of the complexes, which increase the fracture tolerance and reduce the tensile stress. The similar explanation has been reported in our previous articles (*Nat. Chem.* **2016**, 8, 618-624). The sentence of “When the strain speed increases, less time is allowed for the ligand exchange processes and re-formation of the complexes, which reduce the fracture tolerance and increase the tensile stress” has been added in the manuscript.

- In Fig.2C, the Zn-bimcp-PDMS demonstrates excellent stretchability, even with 2400% strain at a loading rate of 10 mm/min, does it restore its initial configuration given enough time? Does the hysteresis also depend on the loading rate? How does the hysteresis change with loading cycles? The reviewer suggests conducting stress-strain curves with multiple cycles for each strain rate.

Response: Thank you for your suggestions.

- 1) The stretched $\text{Zn}(\text{Hbimcp})_2$ -PDMS polymer will not restore its configuration even given enough time. The configuration of stretched polymer has been tracked by camera with the interval of 1 day. As shown in Figure R21, there are no obvious changes in length and shape after 1 day. The reason why the stretched polymer could keep its configuration is that there is rapid ligand exchange processes which fixes the shape after stretching;

Figure R21. The configuration keeping property of the stretched Zn(Hbimcp)₂-PDMS polymer.

2) Yes. The hysteresis depends on the loading rate and loading cycles. When the loading rate was decreased, a higher residual strain and more obvious hysteresis were observed (Figure R22). When increasing the loading cycles, the residual strain of the stretched polymer will increase, and the area of the hysteresis loop will decrease (Figure R23). We have added these data in the revised manuscript;

Figure R22. Cyclic strain-stress curves of the Zn(Hbimcp)₂-PDMS film with different strain rate.

Figure R23. Cyclic strain-stress curves of the $\text{Zn}(\text{Hbimcp})_2\text{-PDMS}$ film with multiple cycles and different strain rate. a, 100 mm min^{-1} . b, 50 mm min^{-1} . c, 25 mm min^{-1} . d, 10 mm min^{-1} .

- In the self-healing experiment, do the authors press the damaged parts together to make them contact intimately? Are there gaps between the healing interface that hinder healing?

Response: Yes, before the self-healing experiment, we need to press the damaged parts to make them contact, but no additional pressure was required during the self-healing process. The gaps between the healing interfaces will hinder healing, because the polymer has high mechanical properties and the fluidity of the polymer is not very good.

- In the coating of sponge, what is the typical thickness of the coating? How does the quality fraction of coated Zn-bimcp-PDMS influence the coating thickness?

Response:

The typical thickness of the coating (d) was calculated using the following equation:

$$d = \frac{\frac{m_p}{\rho_p}}{m_b \cdot S_b} = \frac{m_p}{m_b} \cdot \frac{1}{S_b \cdot \rho_p} = \frac{wt\%}{100 \cdot S_b \cdot \rho_p}$$

Where m_p is the mass of coated polymer, ρ_p is the density of coated polymer, m_b is the mass of blank sponge, S_b is the apparent surface area of blank sponge which is estimated using BET methods, $wt\%$ is the weight ratio of the coated polymer. The density of the Zn(Hbimcp)₂-PDMS polymer ρ_p was 1.07 g cm⁻³ and apparent surface area of blank sponge S_b in this work was estimated as 1.22 m² g⁻¹.

The quality fraction of coated Zn(Hbimcp)₂-PDMS influence significantly on the coating thickness. According to this equation, the thickness of the coatings with different quality fraction of Zn(Hbimcp)₂-PDMS were determined in Table R3 as followed.

Table R3. Characterization of blank sponge and composite sponge

BET surface area of blank sponge (S_b , m ² g ⁻¹)	Density of polymer (ρ_p , g cm ⁻³)	Weight ratio (wt%)	Typical thickness (d , $\mu\text{m}=10^{-6}$ m)
1.22	1.07	20	0.153
		40	0.306
		60	0.459
		80	0.613
		100	0.766

- On line 195, “module” should be “modulus”, please carefully check typos in the manuscript.

Response: Thank you for pointing this out. The mistake has been corrected.

REVIEWERS' COMMENTS:

Reviewer #1 (Remarks to the Author):

The authors have done a decent work on addressing concerns of Reviewer #1. Additional studies of 1) the self-healing properties of Zn₂-bimcp₂-PDMS (Zn:bimcp-PDMS=10:10), and 2) energy absorbing experiments (the effect of compression speed and the duration time at full compression) well-supported the conclusion. The reviewer think this revised paper could be considered for the publication in Nature Communications.

Reviewer #2 (Remarks to the Author):

The authors addressed the comments raised by the reviewers.

One comment:

The model explaining the force spectroscopy is reasonable (Figure R19) if there are free ligands within the chain. The ratio of zinc to ligand was 1:2 in the Synthesis, which would be contradictory to this finding.

One Suggestion:

Table R1 (comparison with the state of the art) should be added to the supporting info. This table provides valuable Information and would be a pity if this table appeared only for the reviewers.

Reviewer #3 (Remarks to the Author):

The authors have addressed all comments that the reviewer raised. The manuscript is accepted to publish.

Reviewer #1 (Remarks to the Author):

The authors have done a decent work on addressing concerns of Reviewer #1. Additional studies of 1) the self-healing properties of Zn₂-bimcp₂-PDMS (Zn:bimcp-PDMS=10:10), and 2) energy absorbing experiments (the effect of compression speed and the duration time at full compression) well-supported the conclusion. The reviewer thinks this revised paper could be considered for the publication in Nature Communications.

Response: Thank you for your positive comments.

Reviewer #2 (Remarks to the Author):

The authors addressed the comments raised by the reviewers.

Response: Thank you for your positive comments.

One comment: The model explaining the force spectroscopy is reasonable (Figure R19) if there are free ligands within the chain. The ratio of zinc to ligand was 1:2 in the Synthesis, which would be contradictory to this finding.

Response: We thank the reviewer for the insightful comment. The sample for force spectroscopy measurement is different from that for mechanical and self-healing studies. That is why the ratio of zinc to ligand seems contradictory. As mentioned in the experimental details, we used a diluted solution of Hbimcp-PDMS and ZnCl₂ in the force spectroscopy measurement to avoid the formation of three-dimensional networks because only polymer chain folded (instead of three-dimensional crosslinking) through coordination bonds can be stretched via AFM cantilever. The results from force spectroscopy can clearly indicate that the single-chain of Zn(Hbimcp)₂-PDMS can be unfolded and quickly refolded thus evidencing the dynamic feature of coordination bonds, but can not fully reveal the complicated situation in three dimensional crosslinked polymer films.

One Suggestion:

Table R1 (comparison) with the state of the art) should be added to the supporting

info. This table provides valuable Information and would be a pity if this table appeared only for the reviewers.

Response: The Table R1 has been added to the supplementary information as Supplementary Table 2.

Reviewer #3 (Remarks to the Author):

The authors have addressed all comments that the reviewer raised. The manuscript is accepted to publish.

Response: Thank you for your positive comments.